# Oxygen-tolerant $CO_2$ electroreduction over covalent organic frameworks via photoswitching control oxygen passivation strategy

Hong-Jing Zhu[1,2,3], Duan-Hui Si[1,2,3], Hui Guo[1,2,3], Ziao Chen[1,2,3], Rong Cao [1,2,3] & Yuan-Biao Huang [1,2,3] ✉

The direct use of flue gas for the electrochemical $CO_2$ reduction reaction is desirable but severely limited by the thermodynamically favorable oxygen reduction reaction. Herein, a photonicswitching unit 1,2-Bis(5′-formyl-2′-methylthien-3′-yl)cyclopentene (DAE) is integrated into a cobalt porphyrin-based covalent organic framework for highly efficient $CO_2$ electrocatalysis under aerobic environment. The DAE moiety in the material can reversibly modulate the $O_2$ activation capacity and electronic conductivity by the framework ring-closing/opening reactions under UV/Vis irradiation. The DAE-based covalent organic framework with ring-closing type shows a high CO Faradaic efficiency of 90.5% with CO partial current density of $-20.1 \, mA \, cm^{-2}$ at $-1.0 \, V$ vs. reversible hydrogen electrode by co-feeding $CO_2$ and 5% $O_2$. This work presents an oxygen passivation strategy to realize efficient $CO_2$ electroreduction performance by co-feeding of $CO_2$ and $O_2$, which would inspire to design electrocatalysts for the practical $CO_2$ source such as flue gas from power plants or air.

The excessive utilization of fossil fuels has led to substantial increase in the atmospheric concentration of carbon dioxide ($CO_2$), resulting in severe challenges such as global climate fluctuation and carbon balance disruption[1,2]. To mitigate these problems, a variety of methods including electrochemical[3,4], photochemical[5,6], and thermochemical[7] reactions have been explored to convert $CO_2$ into value-added products. Among those methods, the electrocatalytic $CO_2$ reduction reaction ($CO_2RR$) driven by renewable energy is a promising and sustainable strategy to reduce the $CO_2$ concentration, and facilitates the production of value-added fuels and chemicals[8–10]. However, the majority of $CO_2RR$ studies were based on pure $CO_2$ streams, which required additional gas separation systems and extra energy consumption[11]. In practical circumstance, the feedstock gas for $CO_2RR$

should be sourced from a real $CO_2$-emitting sources, such as coal-fired gases ($O_2$, ~5%; $CO_2$, ~15%; $N_2$, ~77% and impurities). The direct utilization of a real $CO_2$-emitting sources can avoid the additional energy input and tedious procedure required for $CO_2$ capture and enrichment[11,12]. Nevertheless, the low $CO_2$ concentration and the presence of $O_2$ in the real circumstance usually led to poor selectivity and low energy efficiency for the $CO_2RR$.

This phenomenon arises because the oxygen reduction reaction (ORR) is apparently more thermodynamically favored than the $CO_2RR$ (e.g., $E_{CO_2/CO} = -0.52 \, V$ vs. standard hydrogen electrode (SHE); $E_{O_2/H_2O} = 1.23 \, V$ vs. SHE) under aerobic conditions[13–16]. Moreover, the associated hydrogen evolution reaction (HER) in aqueous electrolyte is preferred to happen at low concentration of $CO_2$[17]. In the past few

[1]State Key Laboratory of Structural Chemistry, Fujian Institute of Research on the Structure of Matter, Chinese Academy of Sciences, 350002 Fuzhou, PR China. [2]Fujian Science & Technology Innovation Laboratory for Optoelectronic Information of China, 350108 Fuzhou, PR China. [3]University of Chinese Academy of Science, 100049 Beijing, PR China. ✉e-mail: ybhuang@fjirsm.ac.cn

years, only a few $CO_2$RR related works under aerobic environment have been explored, such as PIM-CoPc/CNT coated with a microporosity polymer[18] and DEA-$SnO_x$/C modified by diethanolamine[19], which were based on $CO_2$ enrichment strategy. These electrocatalysts typically involved complex electrode preparation process and were unable to completely inhibit the occurrence of ORR. Compared with the physical $CO_2$ adsorption strategy, the inhibition of the more thermodynamically favored reaction ORR in the $CO_2$RR under aerobic environment would be more attractive, but this strategy has not been reported. Therefore, it is necessary to develop new strategies to prepare $O_2$-tolerant electrocatalysts that would facilitate efficient $CO_2$RR with high activity and selectivity under aerobic environment and clarify the inhibition mechanism of ORR.

The key point to suppress the occurrence of the ORR in the $CO_2$RR under aerobic environments is passivation of oxygen and simultaneously promotion of $CO_2$ activation. As we know, the diarylethene (DAE) molecular switches undergo interconversion between their open and closed forms upon exposure to visible and UV light, respectively[20,21]. The structural differences between the states result in a large difference in properties, e.g., electronic energy levels[22], photodynamic therapy[23], biochemical reactivity[24], and photonic devices[25]. Furthermore, the open and close switching structures of DAE lead to the large differences of the energies transfer between $O_2$ and active site[26–28]. Upon the open form of DAE, energy transfer occurs from the active site to $O_2$ ($O_2$ activation), whereas in the close form of DAE, energy transfer does not occur between active site and $O_2$ ($O_2$ passivation)[29]. From these perspectives, the introduction of DAE into the electrocatalytic system would inhibit ORR and promote $CO_2$RR under aerobic environment by controlling the open and close switching structures of DAE.

As one kind of important porous materials, crystalline covalent organic frameworks (COFs), covalently linked by functional organic building blocks[30–32], have demonstrated great potential in various application, including gas storage and separation[33,34], catalysis[35,36], optoelectronics[37], drug delivery[38], and electrochemical for clean energy storage[39,40]. Particularly, their structural diversity, tunable pores and stable frameworks make them compelling catalysts for the $CO_2$RR[14,41,42]. For example, cobalt porphyrins have been precisely integrated into the pre-designed structures of COFs, demonstrating highly selective $CO_2$RR towards the production of CO[43–45]. However, now these COFs only work in pure $CO_2$ streams due to the catalytic activity of the metal porphyrin moieties in the ORR[46]. To the best of our knowledge, there is a lack of reports on utilizing COFs for electrocatalytic $CO_2$RR under aerobic conditions. Therefore, it is urgent to develop new strategies to suppress ORR and design highly efficient $O_2$-tolerant COFs catalysts for the $CO_2$RR in the co-feeding $CO_2$ and $O_2$.

In this work, the photoswitching DAE was introduced into cobalt porphyrin-based COFs to inhibit the ORR, and thus improving the efficiency of the $CO_2$RR under aerobic condition by oxygen passivation strategy. As shown in Fig. 1, 1,2-Bis(5'-formyl-2'-methylthien-3'-yl) cyclopentene in an open form (open-DAE) was firstly installed into a two-dimensional cobalt porphyrin-based COF (denoted as open-DAE-BPy-CoPor) for the highly efficient $CO_2$RR under aerobic condition by modulating the open-DAE to close form (denoted as close-DAE-BPy-CoPor) upon irradiated with UV. For comparison, the parent BPy-CoPor was also prepared by reaction of 2,2'-bipyridine-5,5'-dicarbaldehyde (BPy) and 10,15,20-tetrakis(4-aminophenyl)-porphinatocobalt (Co-TAPP)[47]. Due to the oxygen passivation capability and high electronic conductivity of close-DAE, the close-DAE-BPy-CoPor possess superior activity and selectivity for the $CO_2$RR toward CO by co-

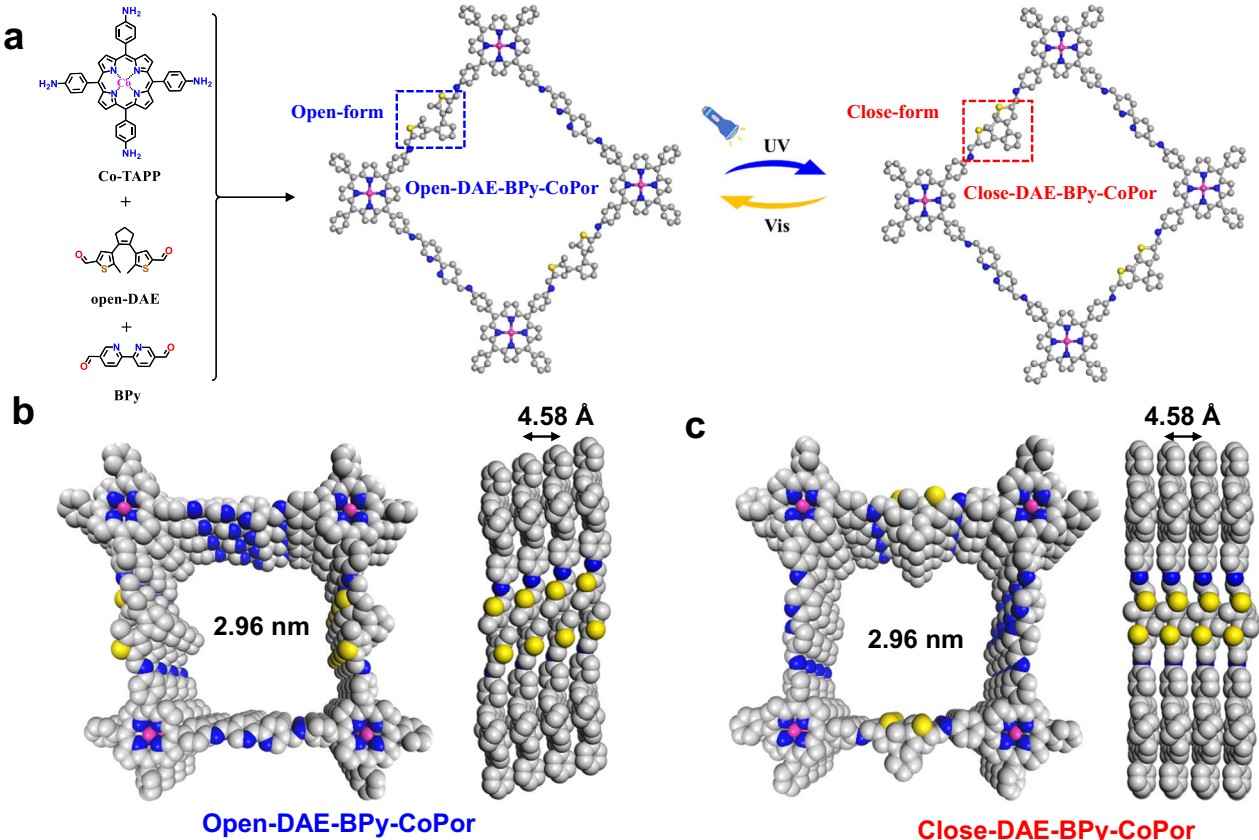

**Fig. 1 | The schematic illustration structures of open-DAE-BPy-CoPor and close-DAE-BPy-CoPor. a** Synthetic route to open-DAE-BPy-CoPor and close-DAE-BPy-CoPor. Simulated structures: Top and side views of **b** open-DAE-BPy-COF and **c** close-DAE-BPy-CoPor, Co magenta, N blue, C gray, S yellow, for clear clarity the H atoms were omitted.

feeding $CO_2$ and 5% $O_2$ with a high $FE_{CO}$ of 90.5% at −1.0 V vs. RHE, which is almost 3.5-fold and 1.2-fold higher than BPy-CoPor ($FE_{CO}$ = 25.9%) and open-DAE-BPy-CoPor ($FE_{CO}$ = 74.9%), respectively. Besides, the CO partial current density ($j_{CO}$) of close-DAE-BPy-CoPor can reach up to −20.1 mA cm$^{-2}$, which is higher than those of BPy-CoPor ($j_{CO}$ = −7.36 mA cm$^{-2}$) and open-DAE-BPy-CoPor ($j_{CO}$ = −15.4 mA cm$^{-2}$) at the same potential. Meanwhile, the DFT calculations reveal that the $O_2$-toleration of close-DAE-BPy-CoPor is benefited from its decreased $O_2$ activation capacity and the higher *OOH formation energy, thereby making the oxygen passivation and suppressing the occurrence of the ORR in the $CO_2$RR under aerobic condition. Moreover, the transition from close-DAE to Co-TAPP demonstrates efficient electron transfer capacity, thus the electron circuit could easily transfer to the Co active sites of Co-TAPP via the close-DAE and a larger current density is obtained.

## Results

### The synthesis and characterization of DAE-BPy-CoPor

Due to the similar molecule size of DAE and BPy (Supplementary Fig. 1), DAE can be introduced into the porphyrin- and bipyridine-based BPy-CoPor by condensation reaction of the three monomers to form the photoswitching open-DAE-BPy-CoPor (Fig. 1). The crystalline structure of open-DAE-BPy-CoPor was confirmed by powder X-ray diffraction (PXRD) analysis with the assistance of Material Studio software[48,49]. The AA stacking mode (Figs. 1 and 2a with blue curve) can greatly reproduce the as-synthesized PXRD patterns (gray curve), as verified by the Pawley refinement with the reasonable weighted-profile R factor ($R_{wp}$, 4.55%) and unweighted-profile R factor ($R_p$, 3.62%). The open-DAE-BPy-CoPor showed intense single peaks at 3.0° and 6.1°, which were assignable to (100) and (020) facets, respectively (Fig. 2a). Notably, open-DAE-BPy-CoPor showed similar PXRD patterns with the parent BPy-CoPor, indicating that the insertion of DAE moiety did not change its topological structure (Supplementary Fig. 2). Therefore, open-DAE-BPy-CoPor crystallizes in the $P1$ space group with the parameters of $a$ = 29.6 Å, $b$ = 29.8 Å, $c$ = 4.58 Å, $\alpha$ = 81.4°, $\beta$ = 73.6° and $\gamma$ = 84.5° (Supplementary Table 1). Thus, one-dimensional (1D) channels were constructed in the well-aligned 2D open-DAE-BPy-CoPor sheets with a theoretical pore size of 2.96 nm, where the distance between adjacent stacking 2D sheet was 4.58 Å (Fig. 1b).

The chemical structure and composition of open-DAE-BPy-CoPor was also confirmed by solid-state $^{13}C$ NMR ($^{13}C$ ssNMR) spectrum and Fourier-transform infrared spectroscopy (FT-IR). The characteristic signal of the C=N imine bond, formed by the condensation reaction, was found at 155.2 ppm in the $^{13}C$ ssNMR spectrum. Besides, the broad peak at 15.5 ppm and 30.3 ppm should be ascribed to the methyl carbons and cyclopentene of the DAE units (Fig. 2b). The FT-IR showed the complete disappearance of the N-H in the range 3400–3100 cm$^{-1}$ for Co-TAPP and C=O at 1600 cm$^{-1}$ for open-DAE, and the appearance of new stretching vibration band of imine bond at 1625 cm$^{-1}$ (Supplementary Fig. 3). All these results confirmed the successful synthesis of the imine-linked open-DAE-BPy-CoPor. Thermogravimetric analysis (TGA) disclosed the open-DAE-BPy-CoPor has a good thermostability up to above 320 °C (Supplementary Fig. 4).

The porosity of open-DAE-BPy-CoPor and BPy-CoPor were assessed by $N_2$ sorption measurements at 77 K. As shown in Fig. 2c, the open-DAE-BPy-CoPor has a higher $N_2$ adsorption uptake with a larger Brunauer–Emmett–Teller (BET) surface area (899.4 m$^2$ g$^{-1}$) in comparison with that of BPy-CoPor (814.6 m$^2$ g$^{-1}$, Supplementary Fig. 5). Furthermore, the typical-IV adsorption isotherm curves observed for open-DAE-BPy-CoPor and BPy-CoPor indicated their mesoporous characters, as revealed by the measured pore size of 2.96 nm (Fig. 2c and Supplementary Fig. 5), which was consistent with the simulated results (Fig. 1b). The high porosity for open-DAE-BPy-CoPor endowed its larger $CO_2$ adsorption uptakes (21.5 cm$^3$ g$^{-1}$ at 298 K) compared to those of BPy-CoPor (19.0 cm$^3$ g$^{-1}$ at 298 K, Supplementary Fig. 6).

Scanning electron microscopy (SEM) and transmission electron microscopy (TEM) were preformed to characterize the morphologies of open-DAE-BPy-CoPor and BPy-CoPor. As shown in Fig. 2d and Supplementary Fig. 7b–d, open-DAE-BPy-CoPor was composed of nanospheres with ~450 nm in diameter by self-assembled small rectangular sheet-shaped crystals, which was similar to the BPy-CoPor (Supplementary Fig. 7a–c). The atomic force microscopy (AFM) images further confirmed their layered structures by griding and high-frequency sonication at room temperature, following a similar procedure to the working electrode preparation. The AFM images showed a thickness of about 1.62 nm, corresponding to three- to four-layer of open-DAE-BPy-CoPor (Supplementary Fig. 8). The structural characteristics of open-DAE-BPy-CoPor was visualized by high resolution transmission electron microscopy (HRTEM). As shown in Fig. 2e, f, the symmetrical diffraction spots and the lattice fringes of ca. 2.50 nm demonstrated the high crystallinity of open-DAE-BPy-CoPor, which was close to the pore size of the simulated framework. Energy-dispersive X-ray spectroscopy (EDX) analysis revealed that Co, S, N and C elements were uniformly distributed over open-DAE-BPy-CoPor (Fig. 2h–l), but no S element was detected in the BPy-CoPor sample. The metal valence states and elemental composition of open-DAE-BPy-CoPor and BPy-CoPor were further detected by X-ray photoelectron spectroscopy (XPS), which further revealed the presence of C, N, S and Co elements in open-DAE-BPy-CoPor (Supplementary Fig. 9). In the XPS spectrum of Co $2p$ for open-DAE-BPy-CoPor, the peaks at 780.2 eV (Co $2p_{3/2}$) and 795.67 eV (Co $2p_{1/2}$) correspond to the +2 state of the cobalt center[14,50], which was same with that of BPy-CoPor (Supplementary Fig. 10). Besides, the analysis of C $1s$, N $1s$ and S $2p$ were plotted in the supplementary information (Supplementary Figs. 9 and 10). The inductively coupled plasma optical emission spectrometry (ICP-OES) revealed that open-DAE-BPy-CoPor and BPy-CoPor have similar Co contents with 4.74 wt.% and 5.36 wt.%, respectively (Supplementary Table 2).

To further explore the local coordination structure of the cobalt species in the open-DAE-BPy-CoPor, X-ray absorption spectroscopy at the Co $K$-edge X-ray absorption near-edge structure (XANES) was conducted (Fig. 3). As shown in Fig. 3a, the Co $K$-edge XANES of open-DAE-BPy-CoPor displayed a pre-edge peak at 7715.3 eV, which was located between CoO and Co foil, implying the positive charged Co was between Co(0) and Co(II). Furthermore, the peak at 7715.3 eV was recognized as 1 s → 4p$_Z$ shake-down transition, which was a fingerprint of the Co-N$_4$ square-planar structure (around 7716 eV)[51]. The Co $K$-edge extended X-ray absorption fine structure (EXAFS) curves for open-DAE-BPy-CoPor showed a main signal at 1.43 Å assigned to the Co-N scattering path, which was similar to that of the molecule [5,10,15,20-tetrakis(4-cyanophenyl) porphyrinato]-Co (Co-TPPCN, 1.47 Å)[52]. The peaks and scattering paths for open-DAE-BPy-CoPor were different from those of Co foil, CoO and Co$_3$O$_4$, further revealing Co-N$_X$ species were predominated in the open-DAE-BPy-CoPor. In addition, the wavelet transform (WT) has been added to prove that there was no CoO species existed in open-DAE-BPy-CoPor. As shown in Supplementary Fig. 11, compared to the WT contour plots of CoO with the feature of Co-Co coordination, the band corresponding to Co-N shown in the WT contour plots of open-DAE-BPy-CoPor and Co-TPPCN without the feature of Co-Co coordination, indicating the absence of CoO species in open-DAE-BPy-CoPor. The EXAFS fitting results revealed that the coordination number of Co species in open-DAE-BPy-CoPor was calculated to be 4.3 (Fig. 3c and Supplementary Table 3). Thus, the above results clearly suggested that the Co atom in the open-DAE-BPy-CoPor was located in a Co-N$_4$ square-planar coordination geometry and the Co porphyrin structure was retained.

The $O_2$ toleration and electrical conductivity would be highly dependent on the switching the close and open form of DAE. Treating the open-DAE-BPy-CoPor with UV light for 3 h to drive the photocyclization of open-DAE and obtain the close-DAE-BPy-CoPor. The

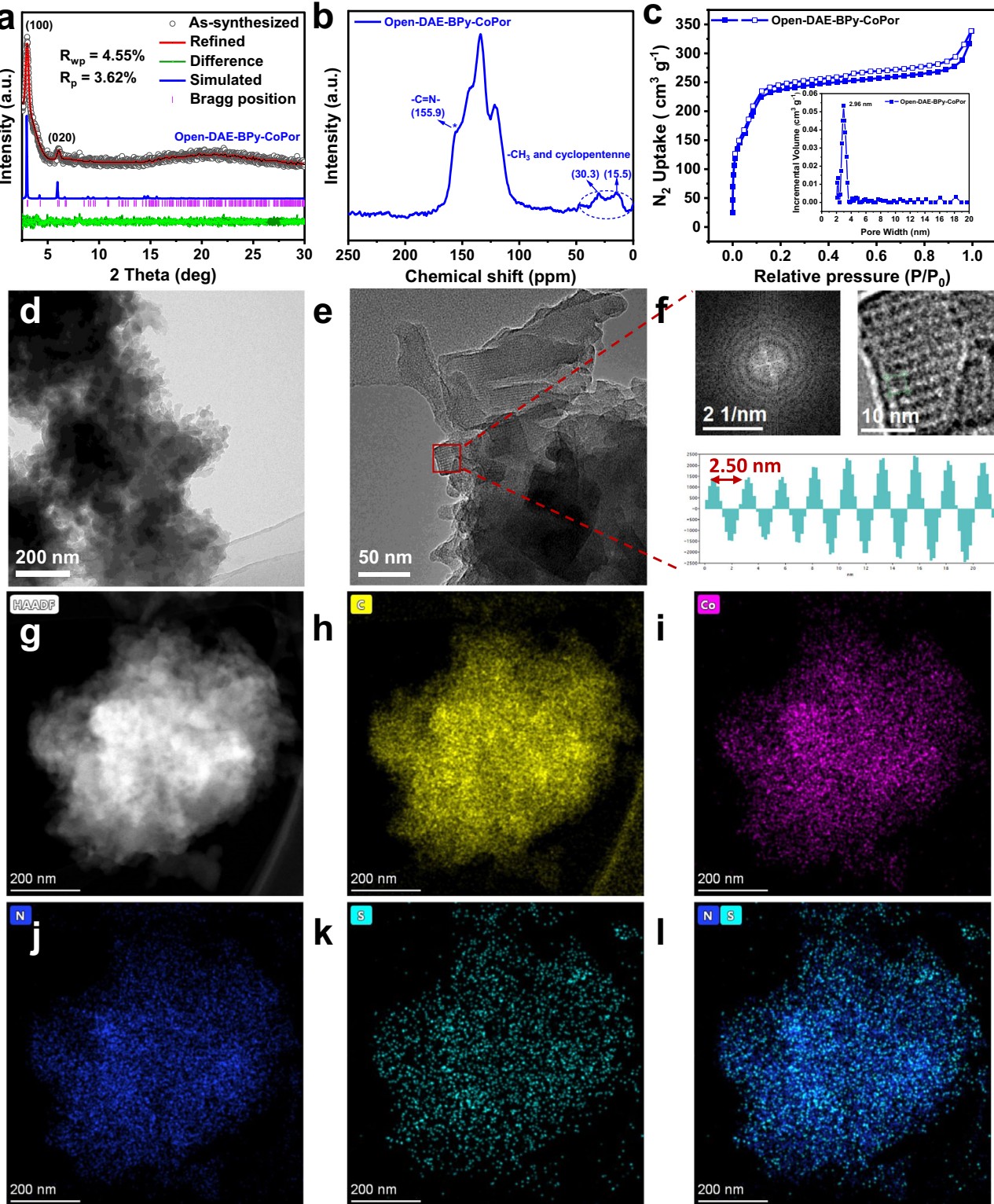

**Fig. 2 | The structure and characterization of open-DAE-BPy-CoPor.**
**a** Comparison of the experimental powder X-ray diffraction pattern with simulated PXRD patterns. **b** Solid-state ¹³C NMR spectrum. **c** N₂ sorption isotherms (Inset: pore width distribution). **d** Transmission electron microscopy and **e** high resolution transmission electron microscopy images. **f** Fast Fourier transformation, high resolution transmission image and lattice distance. **g** Aberration-corrected high-angle annular dark-field scanning transmission electron microscopy image and **h**–**l** energy-dispersive X-ray spectroscopy elemental mapping.

PXRD revealed that the close-DAE-BPy-CoPor was very similar to open-DAE-BPy-CoPor, indicating the photoswitching of DAE moiety did not change its periodic structure and connection mode (Fig. 4a). XPS was performed to monitor the changes in the S $2p$ region of DAE due to the different binding energies of the two isomers[23,26,53]. As shown in Fig. 4c, d, the peak at 164.3 eV was ascribed to the S $2p_{3/2}$, corresponding to the open form of DAE for open-DAE-BPy-CoPor. A positive shift of about 0.4 eV was observed after irradiation by UV light, of which the S $2p_{3/2}$

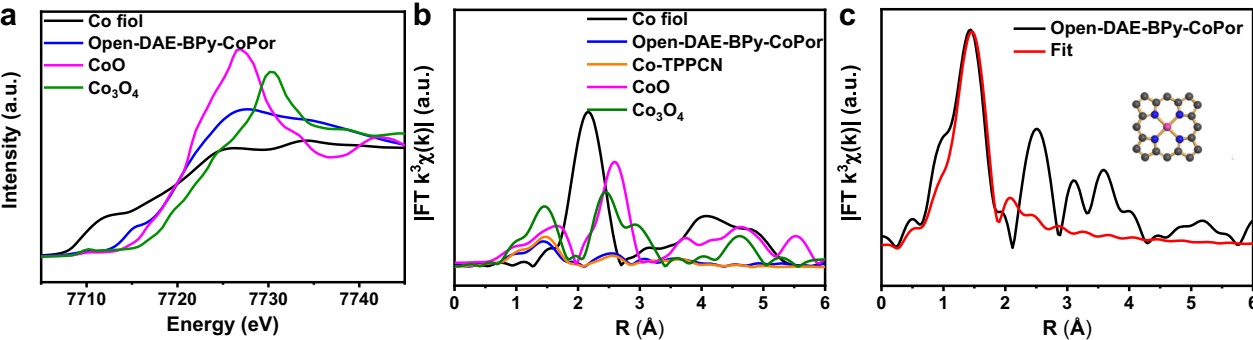

**Fig. 3 | The local coordination structure. a** Co *K*-edge of X-ray absorption near-edge structure spectra of open-DAE-BPy-CoPor, Co foil, CoO, and Co₃O₄. **b** Co *K*-edge of EXAFS spectra of open-DAE-BPy-CoPor, Co foil, Co-TPPCN, Co₃O₄ and CoO. **c** The extended X-ray absorption fine structure fitting curves of open-DAE-BPy-CoPor.

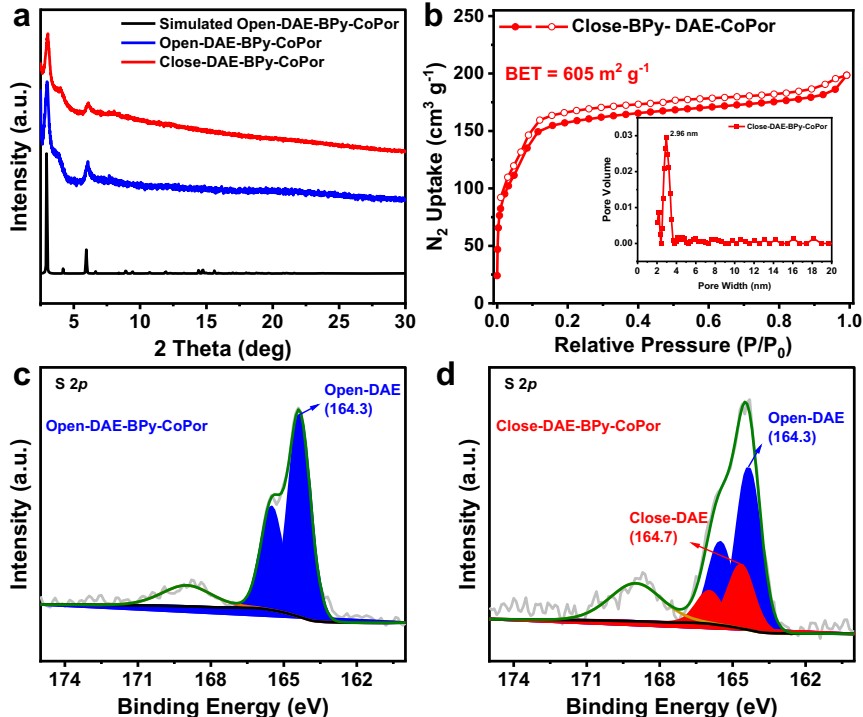

**Fig. 4 | The characterization of close-DAE-BPy-CoPor. a** Comparison of the PXRD patterns of open-DAE-BPy-CoPor and close-DAE-BPy-CoPor. **b** The N₂ sorption isotherms and pore size distribution of close-DAE-BPy-CoPor. **c**, **d** the X-ray photoelectron spectroscopy of S 2*p* region for open-DAE-BPy-CoPor and close-DAE-BPy-CoPor, experimental data, gray line; fitting curve, green line.

peak at 164.7 eV was attributed to the close form of DAE in the close-DAE-BPy-CoPor. The XPS spectrum of C, N and Co elements in close-DAE-BPy-CoPor also showed in Supplementary Fig. 12. After UV irradiation, a growth in the peak area corresponding to the S 2*p* region for close-DAE-BPy-CoPor and a simultaneous decrease in open-DAE-BPy-CoPor were recorded, demonstrating the occurrence of 30% photocyclization for DAE units. Besides, the photocyclization efficiency of DAE was influenced by the duration of UV irradiation exposure, the XPS analysis of close-DAE-BPy-CoPor following 1 h and 5 h after irradiation by UV light revealed the occurrence of 6.3% and 32% photocyclization, respectively (Supplementary Fig. 13). Therefore, considering the constraints of economic and time limitations, we have opted to set the photocyclization time for close-DAE-BPy-CoPor at 3 h of UV irradiation. Then, the changes in the solid-state UV-Vis spectra of close-BPy-DAE-CoPor and open-BPy-DAE-CoPor were further investigated. However, the dark close-DAE-BPy-CoPor and open-DAE-BPy-CoPor (Supplementary Fig. 14) displayed overlapping peaks in the region of 508–633 nm with the close-DAE and the Co-TAPP moieties

(Supplementary Fig. 15b), which made it difficult to detect the photocyclization reaction of the DAE parts in the COF. Nevertheless, the photocyclization of the open-DAE monomer can be detected in the solid-state UV-Vis. As shown in Supplementary Fig. 15a, compared with the spectrum of the monomer open-DAE, a new peak at 640 nm appeared in that of the close-DAE, which indicated the formation of the closed state of the cyclohexadiene moiety in the close-DAE-BPy-CoPor. Furthermore, the Co K-edge XANES profile of close-DAE-BPy-CoPor exhibited a similar wave feature with that of open-DAE-BPy-CoPor, indicating that the Co center valence of close-DAE-BPy-CoPor similar with open-DAE-BPy-CoPor. Notably, compared with open-DAE-BPy-CoPor, the absorption peak of close-DAE-BPy-CoPor at around 7715 eV shifted to the lower-energy side. This shift implied a slightly lower Co oxidation state in close-DAE-BPy-CoPor than open-DAE-BPy-CoPor, indicating the presence of more electrons in the Co center of close-DAE-BPy-CoPor. Moreover, the Fourier-transformed Co K-edge EXAFS spectrum for open-DAE-BPy-CoPor showed a dominant peak at 1.44 Å, which could be attributed to the Co-N bond. Compared with open-

DAE-BPy-CoPor, the intensity of Co-N peak for close-DAE-BPy-CoPor was negatively shifted (Δ = 0.02 Å), indicating a reduced bond length of Co-N in close-DAE-BPy-CoPor (Supplementary Fig. 16b). The EXAFS fitting results further proved that the bond length of Co-N in close-DAE-BPy-CoPor (1.93 Å) was shorter than that of Co-N in open-DAE-BPy-CoPor (1.95 Å) (Supplementary Table 3). The contraction of the Co-N bond length can facilitate in transfer electrons to the Co center[54,55], indicating that the Co center electrons of close-DAE-BPy-CoPor was more than open-DAE-BPy-CoPor. Furthermore, the Co K-edge XANES profile of DAE-BPy-CoPor under varying UV irradiation time (1 h, 3 h and 6 h) also exhibited a similar wave feature with that of open-DAE-BPy-CoPor. However, the absorption peak consistently showed a gradual shift to the lower-energy side as the UV irradiation time increased (Supplementary Fig. 17a). Moreover, in comparison to open-DAE-BPy-CoPor, the EXAFS spectra of close-DAE-BPy-CoPor revealed a consistent negative shift in the Co-N peak as the duration UV irritation varies (Supplementary Fig. 17b). These results suggested that the electronic structures of the Co centers in DAE-BPy-CoPor were affected by the photocyclization efficiency of DAE monomers. It indicated that the presence of more electrons in the Co center of DAE-BPy-CoPor with the increase of the degree of cyclization, which would facilitate in the activation and reduction of $CO_2$.

After photocyclization, the close-DAE-BPy-CoPor still has a high $N_2$ adsorption uptake with a large BET surface area of 605.2 m$^2$ g$^{-1}$ (Fig. 4b), which was slightly lower than that of open-DAE-BPy-CoPor ($S_{BET}$ = 899.4 m$^2$ g$^{-1}$, Fig. 2c and Supplementary Fig. 18). The strong charge delocalization in close-DAE moieties can endow close-DAE-BPy-CoPor with higher electronic conductivity than open-DAE-BPy-CoPor. As expected, the electrical conductivity of close-DAE-BPy-CoPor has an order of magnitude improvement (2.55 × 10$^{-8}$ S m$^{-1}$) than open-DAE-BPy-CoPor (3.40 × 10$^{-9}$ S m$^{-1}$) and BPy-CoPor (6.60 × 10$^{-9}$ S m$^{-1}$) (Supplementary Fig. 19), which would facilitate in the electron transfer to the Co active sites during the $CO_2$RR. Moreover, the electrochemical impedance spectroscopy also proved that close-DAE-BPy-CoPor has better conductivity than open-DAE-BPy-CoPor and BPy-CoPor (Supplementary Fig. 20). These results suggested that the photonic switchable units DAE played an important role in modulating the electronic structure of COFs, and the higher electron conductivity would enhance the electrocatalytic $CO_2$RR activity.

To evaluate the $O_2$-toleration ability of close-DAE-BPy-CoPor, open-DAE-BPy-CoPor and BPy-CoPor, the well-known reactive oxygen species (eg., $^1O_2$, ·OH and ·OOH) scavenger 1,3-diphenylisobenzofuran (DPBF) was used to detect reactive oxygen species for COFs[56]. Upon the oxidative degradation of DPBF by reactive oxygen species, a degradation in the adsorption of DBPF at λ = 410 nm signifies the generation of reactive oxygen species[28]. The experiment was carried out in the 0.1 M tetra-$n$-butylammonium hexafluorophosphate/acetonitrile (TBAPF$_6$/MeCN) bubbled with $O_2$. The carbon paper with COFs acted as the working electrode (detail experiment shown in Methods section). As shown in Supplementary Figs. 21 and 22, the absorbance at λ = 410 nm showed the largest degradation in the BPy-CoPor (0.16, 19% of DPBF absorption decreased), meanwhile, the close-DAE-BPy-CoPor displayed the weakest degradation (0.093, 12% of DPBF absorption decreased), indicating excellent $O_2$ toleration ability of close-DAE-BPy-CoPor. Besides, the ORR was further as model reaction to evaluate oxygen-activation ability of these COFs. As shown in Supplementary Fig. 23, the close-DAE-BPy-CoPor showed the lowest oxygen activity, characterized by the most negative half-wave potential and the smallest diffusion-limiting current density compared with open-DAE-BPy-CoPor and BPy-CoPor.

## The electrocatalytic $CO_2$RR performances

Above all, the obtained porous close-DAE-BPy-CoPor and open-DAE-BPy-CoPor constructed from Co-TAPP, BPy and DAE with switchable electron transfer ability and oxygen tolerance might serve as promising candidates for $CO_2$RR in the presence the $O_2$. To evaluate their $CO_2$RR performances, the linear sweep voltammetry (LSV) in a pure $CO_2$-saturated 0.5 M KHCO$_3$ aqueous solution were firstly investigated. As shown in Fig. 5a, close-DAE-BPy-CoPor has a more positive onset potential and higher current densities than those of BPy-CoPor and open-DAE-BPy-CoPor in the applied potentials of −0.3 to −1.2 V, which can be attributed to that close-DAE-BPy-CoPor has stronger electron transfer ability than the latter two samples. The GC tests demonstrated that the gas products of the $CO_2$RR were CO and $H_2$ during the investigated potentials. Notably, no liquid product was detected in the $^1$H-NMR (Supplementary Fig. 24). As shown in Fig. 5b, the close-DAE-BPy-CoPor exhibited remarkable Faradaic efficiencies of CO (FE$_{CO}$) (≥ 90%) across the entire potential window from at −0.6 V to −0.9 V (*vs.* RHE), which were larger than those of BPy-CoPor and open-DAE-BPy-CoPor under pure $CO_2$ gas (Supplementary Fig. 25 and Supplementary Table 4). Particularly, the close-DAE-BPy-CoPor achieved almost 100% FE$_{CO}$ at −0.7 V. Furthermore, close-DAE-BPy-CoPor also showed outstanding CO partial current density ($j_{CO}$) and reached −8.47 mA cm$^{-2}$ at −0.9 V, which was 2.0-fold and 1.2-fold higher than that of BPy-CoPor (−4.22 mA cm$^{-2}$) and open-DAE-BPy-CoPor (−6.99 mA cm$^{-2}$), respectively (Fig. 5c). Notably, only $H_2$ as the gas product was detected by the GC over 5,10,15,20-tetrakis (4-aminophenyl)-21H,23H-porphine (H$_2$-TAPP), which indicated the Co acted as the active center in the $CO_2$RR (Supplementary Fig. 26). In addition, we increased the amount of carbon black from 0.5 times (carbon black: electrocatalyst) to 1.5 times, the $j_{CO}$ of close-DAE-BPy-CoPor achieved −38.0 mA cm$^{-2}$ at −1.2 V (Supplementary Fig. 27). Besides, only $H_2$ was obtained on the carbon paper only coated with carbon black in $CO_2$-saturated 0.5 M KHCO$_3$ (Supplementary Fig. 28). Besides, we conducted an in-depth investigation into the performance of close-DAE-BPy-CoPor in a gas diffusion electrode cell (GDE). As shown in Supplementary Fig. 29, the $j_{CO}$ reached −15.0 mA cm$^{-2}$ at −0.9 V in a pure $CO_2$ condition. Additionally, we further study the $CO_2$RR performance of close-DAE-BPy-CoPor in 0.1 M KOH and 1 M KOH. As illustrated in Supplementary Fig. 29c, d, the FE$_{CO}$ of close-DAE-BPy-CoPor was above 90% in a wide potential range (−0.3 V to −1.3 V). Furthermore, the partial current density of CO reached −44.2 mA cm$^{-2}$ at −1.3 V in 0.1 M KOH, which was superior to most reported porphyrin-based catalysts. Furthermore, the FE$_{CO}$ of close-DAE-BPy-CoPor was above 90% in the potential range from −0.3 to −0.5 V, and $j_{CO}$ reached −108.9 mA cm$^{-2}$ at an applied potential of −0.8 V in 1 M KOH. To further understand the intrinsic activity of the close-DAE-BPy-CoPor, the calculation of turnover frequency (TOF) based on the number of metals in the reaction was determined, and the calculated TOF at −0.9 V for close-DAE-BPy-CoPor reached up to 187 h$^{-1}$, which was 2.2-fold and 1.2-fold higher than BPy-CoPor (85 h$^{-1}$) and open-DAE-BPy-CoPor (154 h$^{-1}$) (Supplementary Fig. 30). The superior activity on close-DAE-BPy-CoPor can be reasonably attributed to the strong electron transfer ability when the open-state of DAE was switched to the close-state in the DAE-BPy-CoPor.

The origin of the $CO_2$RR products over close-DAE-BPy-CoPor were ascertained by the isotopic label experiments, in which using $^{13}CO_2$ to trace the carbon sources of CO in KCl or KHCO$_3$ electrolyte. As shown in Supplementary Fig. 31, only the signals at m/z = 29 assigned to $^{13}$CO was detected in $^{13}CO_2$ saturated 0.5 M KCl, and the fragments of $^{13}$CO (m/z = 29) were also observed in 0.5 M KHCO$_3$. The signal at m/z = 28 belonged to $^{12}$CO also detected in $^{13}CO_2$ saturated 0.5 M K$^{12}$HCO$_3$, which was likely originated form the $^{13}CO_2$(g) in equilibrium with H$^{12}CO_3^-$. This phenomenon was also observed in other $CO_2$RR over Cu-based species and single atomic catalysts[57,58]. Those results illustrated that the carbon product CO was originated from the $CO_2$ gas in equilibrium with HCO$_3^-$ in 0.5 M KHCO$_3$ aqueous electrolyte. The long-term stability of close-DAE-BPy-CoPor was studied by chronoamperometric test in a $CO_2$-saturated 0.5 M KHCO$_3$ electrolyte, which showed the corresponding FE$_{CO}$ can be retained (FE$_{CO}$ ≥ 80%) and there was 36.2%

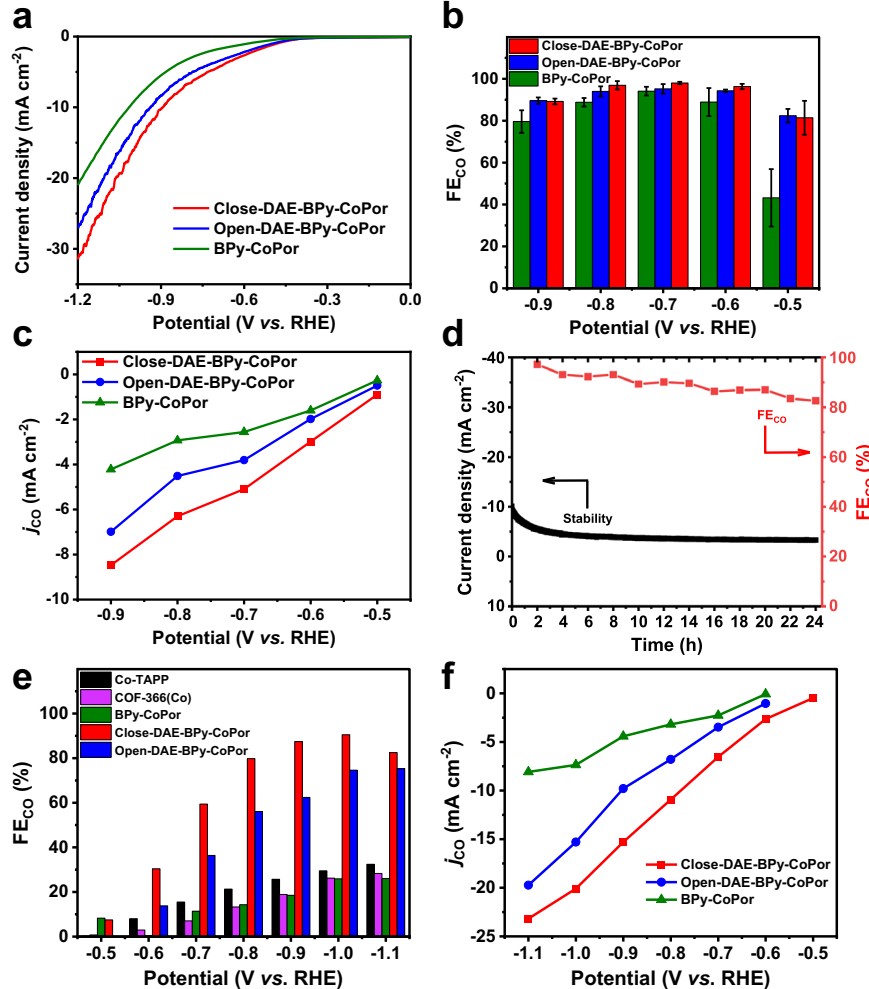

**Fig. 5 | The electrocatalytic CO₂RR performances. a** Linear sweep voltammetry curves, **b** the CO Faradic efficiency (error bars are determined from five replicate trials at different potentials) and **c** the CO partial current density of the BPy-CoPor, close-DAE-BPy-CoPor and open-DAE-BPy-CoPor. **d** Stability of close-DAE-BPy-CoPor at −0.7 V vs. RHE. All the above tests were conducted in a CO₂-saturated 0.5 M KHCO₃ aqueous solution under dark environment. **e** The CO Faradic efficiency of BPy-CoPor, open-DAE-BPy-CoPor, close-DAE-BPy-CoPor, COF-366(Co) and Co-TAPP under aerobic conditions. **f** The CO partial current density of the BPy-CoPor, close-DAE-BPy-CoPor and open-DAE-BPy-CoPor in the co-feeding CO₂ and 5% O₂.

current drop (−5.20 mA cm⁻² drop to −3.32 mA cm⁻²) at −0.7 V after 24 h (Fig. 5d). We have fine-tuned the catalytic potential for close-DAE-BPy-CoPor, enabling a stable current density for 24 h at −0.9 V (vs. RHE), while maintaining a CO selectivity exceeding 80% (Supplementary Fig. 32). The ICP-MS revealed that less than 0.01% of Co atoms leached from close-DAE-BPy-CoPor to the catholyte solution after long-term electrocatalysis. Besides, the HRTEM retained the clear lattice fringes of close-DAE-BPy-CoPor. Additionally, the EDX elemental mapping revealed a uniform distribution of all atoms over its entire framework. Notably, no Co or CoOₓ based nanoparticles were detected after long-term usage, demonstrating the electrochemical stability of close-DAE-BPy-CoPor (Supplementary Figs. 33 and 34) Furthermore, the PXRD pattern and FT-IR of the close-DAE-BPy-CoPor after CO₂RR remained consistent with that of the as-synthesized COF, confirming the structural integrity of the close-DAE-BPy-CoPor (Supplementary Fig. 35).

The remarkable CO₂ reduction activity exhibited by close-DAE-BPy-CoPor in pure CO₂ system encouraged us to study its performance in the CO₂RR under co-feeding CO₂ and 5% O₂. To investigate the role of close-DAE in close-DAE-BPy-CoPor, the COF-366(Co) containing-free DAE that constructed with Co-TAPP and terephthalaldehyde was also prepared. As shown in Fig. 5e, the close-DAE-BPy-CoPor still has high FE_CO values at a wide range of the applied potentials from −0.8 V

to −1.1 V. The optimal FE_CO was up to 90.7% at −1.0 V for close-DAE-BPy-CoPor, which was nearly 2.2-fold, 2.2-fold, 3.5-fold and 1.2-fold higher than those of Co-TAPP (FE_CO = 41.1%, −1.2 V), COF-366(Co) (FE_CO = 41.0%, −1.2 V), BPy-CoPor (FE_CO = 26.0%, −1.1 V) and open-DAE-BPy-CoPor (FE_CO = 75.3%, −1.1 V) (Supplementary Fig. 36). Besides, the j_CO of close-DAE-BPy-CoPor can reach up to −20.1 mA cm⁻² at −1.0 V, which was higher than BPy-CoPor (j_CO = −7.36 mA cm⁻²) and open-DAE-BPy-CoPor (j_CO = −15.4 mA cm⁻²) at the same potential (Supplementary Figs. 37 and 38, Fig. 5f). Besides, the close-DAE-BPy-CoPor also demonstrated excellent CO selectivity in a CO₂ + N₂ feeding environment, but only H₂ was obtained under pure N₂ feeding environment (Supplementary Fig. 39). Under varying concentrations of diluted CO₂ (77%, 50% and 23%, another mixed gas was N₂), the close-DAE-BPy-CoPor produced CO with FE_CO ≥ 90% under 77% CO₂ + 23% N₂, while at a lower CO₂ concentration (23% CO₂ + 77% N₂), the FE_CO of close-DAE-BPy-CoPor can reach 89.2% at −0.6 V, showing the potential of close-DAE-BPy-CoPor for practical applications (Supplementary Fig. 40). To the best of our knowledge, the CO₂RR performance under aerobic environment of close-DAE-BPy-CoPor was one of the several highly selective electrocatalysts (Supplementary Fig. 41). Those data indicated the DAE with close state in close-DAE-BPy-CoPor effectively inhibited the ORR side reaction. In order to probe the deep mechanism of oxygen passivation during CO₂RR, operando ATR-FTIR experiments

were conducted in the 0.5 M KHCO$_3$ under the pure CO$_2$ steams or co-feeding CO$_2$ and O$_2$ (Supplementary Fig. 42). As shown in Supplementary Fig. 43, the band located at 1396 cm$^{-1}$ in the close-DAE-BPy-CoPor and open-DAE-BPy-CoPor spectra under pure CO$_2$ and aerobic environments were assigned to a carboxyl intermediate of *COOH, recognized as the key intermediate for the formation of CO[59,60]. Notably, under aerobic environment, a broad band around ~971 cm$^{-1}$ was observed in the open-DAE-BPy-CoPor (Supplementary Fig. 44), which was associated with the *OOH intermediate of ORR[61]. However, no obvious band around ~971 cm$^{-1}$ was observed in the close-DAE-BPy-CoPor under aerobic environment. Additionally, the operando ATR-FTIR measurement for BPy-CoPor and open-DAE-BPy-CoPor under CO$_2$ + O$_2$ mixed gas showed that the band located at 1398 cm$^{-1}$ in the open-DAE-BPy-CoPor and BPy-CoPor spectra was assigned to a carboxyl intermediate of *COOH (Supplementary Fig. 45). More importantly, in the BPy-CoPor curve, an obviously downward broad band at 1637 cm$^{-1}$ was observed, assigned to the H-O-H bending generation, suggesting a hint of water production during the reaction. Furthermore, the intensity of H-O-H bending band was stronger than *COOH band, which indicated a greater propensity for the occurrence of the ORR rather than CO$_2$RR in BPy-CoPor. These test data indicated that the excellent CO$_2$RR performance of close-DAE-BPy-CoPor in the co-feeding CO$_2$ and O$_2$ benefited from the O$_2$ passivation of close-DAE.

## The DFT calculation and reaction mechanism

The density functional theory (DFT) was further conducted to confirm the above speculation, providing a deeper understanding of the catalytic selectivity and offering insights into the mechanism of CO$_2$RR. The Gibbs free energies curves of CO$_2$RR and HER on the open-DAE-BPy-CoPor and closed-DAE-BPy-CoPor were shown in Fig. 6. The calculation intermediate models of the CO$_2$RR, HER and ORR were presented in the supplementary information from Supplementary Fig. 46 to Supplementary Fig. 51. Firstly, CO$_2$ molecule was adsorbed on the active cobalt center to generate *+CO$_2$ (where * means the active site), which was then activated to form a carboxyl intermediate (*COOH) by the first proton-electron transfer process (Supplementary Data 1–3, 12–14). The *COOH was subsequently by the second proton-electron transfer process converted to *CO which was finally desorbed from the Co site to generate CO (Supplementary Data 4, 5, 15 and 16).

As shown in Fig. 6a, the *COOH formation was the rate-determining step (RDS) of the CO$_2$RR for both of open-DAE-BPy-CoPor and closed-DAE-BPy-CoPor with the free energies of 0.47 eV and 0.43 eV, respectively, which were lower than the free energies of the formation H$_2$ from *H (0.92 eV and 1.05 eV) process for HER (Supplementary Data 6, 7, 17, and 18). Thus, CO$_2$RR would be more easily occurred on open-DAE-BPy-CoPor and closed-DAE-BPy-CoPor than HER in an aqueous electrolyte. Besides, we recalculated the free energies of CO$_2$RR at $U = -0.7$ V, and the overall free energy surface mentioned was exothermic[62,63] (Supplementary Fig. 52). Moreover, the energy for RDS on the close-DAE-BPy-CoPor (0.43 eV) was lower than that of open-DAE-BPy-CoPor (0.47 eV), which clearly showed that the high activity and selectivity CO$_2$RR of close-DAE-BPy-CoPor in the CO$_2$RR, and the Bader charge analysis of the Co active sites further explained it. The Bader charge of the Co active sites of close-DAE-BPy-CoPor and open-DAE-BPy-CoPor were counted. As clearly shown in Fig. 6b, close-DAE-BPy-CoPor showed more charge transfer to CO$_2$ (−1.170) than that of open-DAE-BPy-CoPor (−0.984), which indicated the close-DAE-BPy-CoPor has more charge distribution in the active site compared with the latter sample. Furthermore, the free energy of ORR pathways (Supplementary Fig. 53) and the projected electron density (PDOS) of states analysis of close-DAE-BPy-CoPor and open-DAE-BPy-CoPor further clarify the reason for the different performance of close-DAE-BPy-CoPor under aerobic condition. The formation free energy for the key intermediate *OOH of the ORR on the Co active cite of open-DAE-BPy-CoPor (−0.80 eV) was more negative than that on close-DAE-BPy-CoPor (−0.77 eV), indicating the better activation of open-DAE-BPy-CoPor for *OOH (Supplementary Fig. 53, Supplementary Data 8–11, 19–22). Furthermore, the PDOS analysis indicated that the O$_2$ in the open-DAE-BPy-CoPor was closer to the Fermi level compared to close-DAE-BPy-CoPor catalyst (Fig. 6c) and exhibited the weaker hybridization between O$_2$ and Co active sites in the close-DAE-BPy-CoPor, which was well agreed that the electrons on close-DAE-BPy-CoPor were hardly transfer to O$_2$, thus passivating O$_2$ on close-DAE-BPy-CoPor under aerobic condition (Fig. 5e and Supplementary Fig. 44). The integrated DOS also showed the Co orbital electron occupancy number of close-DAE-BPy-CoPor (7.77) was lower than open-DAE-BPy-CoPor (12.09), indicting Co center of close-DAE-BPy-CoPor has a weak O$_2$ activation ability, thus enhancing the reactivity of CO$_2$RR under

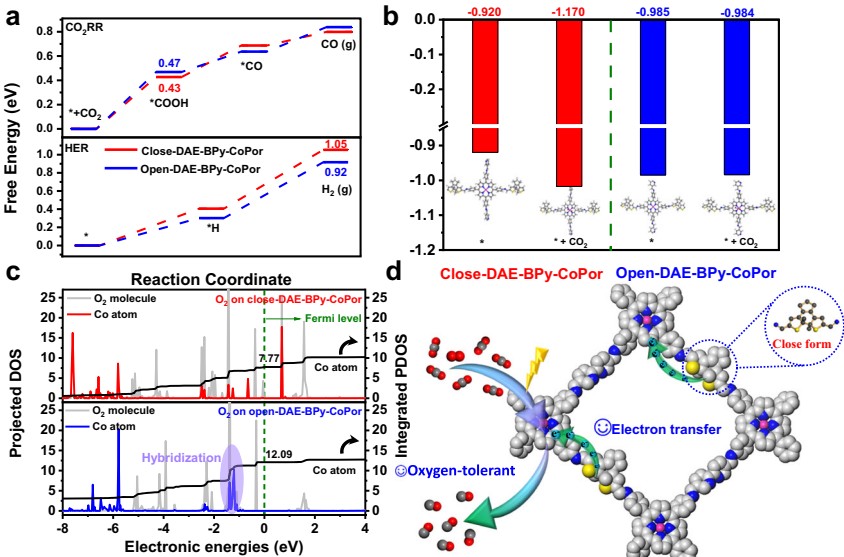

**Fig. 6 | Density functional theory calculations and proposed schematic mechanism. a** Free energy diagrams of close-DAE-BPy-CoPor and open-DAE-BPy-CoPor for CO$_2$RR and HER pathways. **b** The Bader charge analysis of different Co atoms in close-DAE-BPy-CoPor and open-DAE-BPy-CoPor (inset: the calculation intermediate models of the CO$_2$RR). **c** Projected density of states and integrated density of states of adsorption structures of O$_2$ on close-DAE-BPy-CoPor and open-DAE-BPy-CoPor. **d** Proposed schematic mechanism for the CO$_2$RR on close-DAE-BPy-CoPor under aerobic conditions.

aerobic conditions. In general, the excellent $CO_2RR$ performance of the close-DAE-BPy-CoPor under co-feeding $CO_2$ and $O_2$ benefits from the more difficult formation of *OOH and passivating $O_2$.

In summary, the photoswitching built block DAE has been installed into a 2D porphyrin- and bipyridine-based BPy-CoPor to tune the electron transfer rate and oxygen activation ability for the enhanced $CO_2RR$ performance in the pure $CO_2$ gas or under aerobic conditions. Compared with the open-DAE-BPy-CoPor (open-form DAE) and BPy-CoPor (without DAE), the close-DAE-BPy-CoPor showed stronger transfer ability and highest CO formation ability. Thus, the close-DAE-BPy-CoPor has the highest CO selectivity with $FE_{CO}$ close to 100% and largest partial current density at the applied potential of −0.7 V vs. RHE in the pure $CO_2$ gas. Due to the DAE can reversibly modulate the $O_2$ activation capacity by the DAE ring-closing/opening reactions, the close-DAE-BPy-CoPor has a lowest oxygen activation capacity. Thus close-DAE-BPy-CoPor showed superior $CO_2RR$ performance with $FE_{CO}$ up to 90% and higher partial current density under aerobic conditions. The DFT calculations and operando ATR-FTIR experiments illustrated that the excellent $CO_2RR$ performance of close-DAE-BPy-CoPor in co-feeding $CO_2$ and $O_2$ originated from the lower $O_2$ activation ability and higher energy transfer $O_2$ into *OOH (the ORR limiting step). This work sheds a new light on reversibly modulating the oxygen activation ability of $O_2$-tolerant electrocatalysts for the $CO_2RR$ in the presence of $O_2$.

## Methods
### Synthesis of BPy-CoPor
BPy-CoPor was synthesized following previously reported literature with a slight modification[47]. In detail, Co-TAPP (14.5 mg, 0.02 mmol) and BPy (8.5 mg, 0.04 mmol), benzyl alcohol (0.75 mL), o-dichlorobenzene (0.25 mL), and 6 M aqueous acetic acid (0.1 mL) were added in a Pyrex tube (1 × 20 cm in outside diameter × length). The mixture was subjected to sonication for approximately 15 min, followed by flash freezing at 77 K (liquid $N_2$ bath) and degassing to achieve an internal pressure of -100 mTorr. Upon returning to the room temperature, the mixture was heated at 120 °C and allowed to stand undisturbed for 72 h. After filtration, the wet sample was transferred to a Soxhlet extractor, washing with THF and acetone for 24 h respectively. And the product was evacuated at 70 °C under vacuum overnight to obtain the activated sample.

### Synthesis of close-DAE-BPy-CoPor
The close-DAE-BPy-CoPor was formed by exposing dispersion solution (THF) of open-DAE-BPy-CoPor to UV (-365 nm) for 3 h.

### Materials and synthetic procedures
All reagents and chemicals were obtained commercially and used without further purification. Cobalt acetate (Co(OAc)$_2$·H$_2$O) and 2,2′-bipyridine-5,5′-dicarbaldehyde (BPy) were purchased from Alfa Aesar. 1,2-Bis(5′-formyl-2′-methylthien-3′-yl) cyclopentene (open-DAE) wad purchased from Jilin Chinese Academy of Sciences-Yansheng Technology Co., Ltd. Deionized water was supplied with a UPT-I-5T ultra-pure water system (18.25 MΩ cm).

### Characterizations and instruments
Scanning electron microscopy (SEM) images were obtained using a JSM6700-F working at 10 kV. Transmission electron microscope (TEM) images were recorded by a FEIT 20 working at 200 kV. Aberration-corrected high-angle annular dark-field scanning transmission electron microscopy (HAADF-STEM) images and the EDS of samples were performed with a Titan Cubed Themis G2 300 (FEI) high-resolution transmission electron microscope operated at 200 kV. Powder X-ray diffraction (PXRD) patterns were recorded on a Miniflex 600 diffractometer using Cu Kα radiation (λ = 0.154 nm). $N_2$ sorption isotherm and the Brunauer-Emmett-Teller (BET) surface area measurements

were measured using Micromeritics ASAP 2460 instrument. $CO_2$ sorption isotherms were measured using Micromeritics ASAP 2020 instrument. The FT-IR spectra were measured using VERTEX70 (Bruker). X-ray photoelectron spectroscopy (XPS) measurements were performed on an ESCALAB 250Xi X-ray photoelectron spectrometer (Thermo Fisher). XAFS spectra at the Cu K-edge (8979 eV) were measured at the beamline BL14W1 station of the Shanghai Synchrotron Radiation Facility, China. ATR-FTIR experiments were performed on a Nicolet6700 (Thermo Fisher) equipped a liquid nitrogen cooled MCT detector. The gas chromatography measurements were performed on the FULI INSTRUMENTS GC9790 PLUS gas chromatograph (GC) equipped with FID and TCD. The analysis of metal content was measured by inductively coupled plasma atomic emission spectroscopy on an Avio220Max. The isotopic species was purchased from WUHAN NEWWRADAR SPECIAL GAS Co., LTD and the enrichment is 99 atom% of $^{13}C$.

### Electrochemical measurements
All the electrochemical experiments were conducted in an H-type cell featuring two compartments separated by Nafion-117 exchange membrane[64,65]. Each compartment was filled with 0.5 M KHCO$_3$ (70 ml). Using carbon paper as the working electrode, the Ag/AgCl electrode as the reference electrode and Pt foil as the counter electrode to measure the $CO_2RR$ performance of the catalyst. To prepare working electrode, typically, catalyst (5 mg), carbon black (2.5 mg or 7.5 mg) and 5 wt% Nafion (40 μL) were dispersed in isopropanol (460 μL). The mixture was then subjected to sonicatin for 30 min. Subsequently, 50 μL of the resulting ink was deposited onto the surface of carbon paper with an area of 1 cm$^2$ and allowed to dry at room temperature for 12 h. During the process of evaluating $CO_2RR$ performance, the electrolyte solution was purged with high-pure $CO_2$ or $CO_2$ in the presence of $O_2$ for 30 min (pH = 6.8). A mass flow controller was employed to set the $CO_2$ flow rate at 30 sccm. All applied potentials were converted to hydrogen electrode reports using the formula E (vs. RHE) = E (vs. Ag/AgCl) + 0.196 V + 0.059 × pH, without any IR compensation. The LSV curves were obtained with a scan rate of 5 mV/s. The gas products of $CO_2RR$ were detected by gas chromatograph every 10 min, and the liquid products were analyzed by quantitative NMR using dimethyl sulphoxide (DMSO) as an internal standard.

The Faraday efficiency of the gas product was calculated by the following equation:

$$FE = \frac{PV}{T} \times \frac{\upsilon NF \times 10^{-6}(m^3/mL)^{-6}}{I \times 60(s/\min)} \tag{1}$$

$\upsilon$ (vol %): volume concentration of the gas product in the exhaust gas from the cell based on GC data;
$V$: gas flow rate according to flow meter, 30 mL min$^{-1}$;
$I$: total steady-state cell current;
$N$: the electron transfer number for product formation;
$F$: Faradaic constant, 96485 C mol$^{-1}$;
$R$: universal gas constant, 8.314 J mol$^{-1}$ K$^{-1}$;
$P$: one atmosphere, 1.013 × 10$^5$ Pa;
$T$: room temperature, 298.15 K.

## Data availability
This study is available from the corresponding author upon request. The source data underlying Figs. 2a–i, 3a–c, 4a–d, 5a–f, and 6a–c are provided as a Source Data file. Source data are provided with this paper.

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

## Acknowledgements
The work was supported by the National Key Research and Development Program of China (2023YFA1507904 and 2021YFA1501500 to Y.-B.H.), NSFC (U22A20436 and 22071245 to Y.-B.H., 22220102005 and 22033008 to R.C.), and Fujian Science & Technology Innovation Laboratory for Optoelectronic Information of China (2021ZZ103 to R.C.). This research was supported by Open Science Promotion Plan 2023 of CSTCloud (to D.H.S.). The authors thank the beamline BL14W1 station for XAS measurements at the Shanghai Synchrotron Radiation Facility, China.

## Author contributions
H.-J.Z., R.C., and Y.-B.H. conceived the idea. H.-J.Z. designed the experiments, collected and analyzed the data. H.G. and Z.C. assisted with the experiments and characterizations. D.-H.S. performed structure simulations and accomplished the theoretical calculation. H.-J.Z. wrote the manuscript. All the authors contributed to the manuscript preparation.

## Competing interests
The authors declare no competing interests.
