## [Peer Review File · Nature Communications]

REVIEWER COMMENTS

Reviewer #1 (Remarks to the Author):

The authors reported a cobalt porphyrin-based covalent organic framework integrated with 1,2-Bis(5'-formyl-2'-methylthien-3'-yl)cyclopentene (DAE) moiety (DAE-BPy-CoPor), of which the framework shows ring-closing/opening features for modulating the oxygen-passivation behaviors during CO₂RR. However, the characterization and mechanism of the oxygen passivation effect is very insufficient, and the conclusion is too simplistic and crude.

In both CO₂-saturated and CO₂/O₂ co-feeding condition, the close-DAE-BPy-CoPor showed a moderate higher CO partial current density than that of open-DAE-BPy-CoPor, which should be mainly responsible to their different electronic conductivity of the frameworks. Therefore, it is too preliminary to conclude that the slight activity enhancement of close-DAE-BPy-CoPor results from the oxygen-passivation, instead of its higher electronic conductivity.

It is too strange that the CO partial current densities of the catalysts conducted in a CO₂-saturated condition are higher than those gained in the co-feeding CO₂ and 5% O₂, even the competitive ORR occurs.

Actually, the catalytic activity of close-DAE-BPy-CoPor is quite poor, also inferior to most of recently reported catalysts. For example, in CO₂-saturated CO₂RR, close-DAE-BPy-CoPor only showed a low partial current density of -8.5 mA cm⁻² at -0.9 V. Moreover, the stability is low, as both current density and Faradaic efficiency dropped quickly during the stability test.

The ring-closing/opening forms of DAE-BPy-CoPor could lead to the electronic regulation on the CoN₄ centers and thus their different catalytic activity. To identify the electronic difference, the Co K-edge of EXAFS spectra both open-DAE-BPy-CoPor and closing-DAE-BPy-CoPor should be provided.

In short, the main conclusions derived in the manuscript are not sufficiently justified. Consequently, this reviewer does not believe the current manuscript should be considered for publication in Nature Communications.

Reviewer #2 (Remarks to the Author):

Reviewer #3 (Remarks to the Author):

1) Key Results

In this study, Cao, Huang, and co-workers report using a photoswitching unit (DAE) integrated into a cobalt porphyrin-based covalent organic framework (COF) for the CO₂ reduction reaction (CO₂RR) under

an aerobic environment. The authors have identified a significant problem in the CO₂RR reduction strategy, suppressing the oxygen evolution reaction (ORR) while selectively reducing CO₂ to value-added products. The authors have provided adequate experimental evidence for synthesizing the COFs, including spectroscopic techniques such as SEM, TEM, and PXRD. They have also employed XPS to provide local coordination structure of their materials, a significant part of the work, and performed DFT calculations to probe the reaction mechanism.

The electrochemical performance evaluation also constitutes a noteworthy part of the work, especially where all critical variations of the COFs are considered, and proper trends are reflected and discussed accordingly. It is important to note that the close-DAE-COF obtained is stable over 24 hours and can maintain a good Faradaic efficiency (FE) for CO₂RR.

Overall, Cao, Huang, and co-workers have successfully achieved their strategy to make an O₂-tolerant COF for CO₂RR where the efficiency can be controlled by light.

2) Conclusions, claims & additional evidence.

The experimental work presented here does support the mentioned conclusions. However, there are a few methodologies and instrumentation information that are missing, critical for the reproducibility of the results:

- i) Isotopic label experiments. Were the isotopic species procured from commercial sources or synthesized in the lab?
- ii) GC & ICP-MS – The methodology and instrument details are missing in the “Characterizations and Instruments “ section.

The computational part is also missing information central to the reproduction of results, such as:

- iii) The Bader charge on the Co-atoms was counted, but the methodology implemented here is not mentioned; for example, how were the Co-atoms counted in the analysis? (Lines 406-408)
- iv) The computed Cartesian coordinates should be provided in the supporting information.
- v) The authors should consider discussing how the density of states calculations (projected and integrated) were obtained.
- vi) The overall free energy surface mentioned is endothermic, indicating that the reaction would not occur (i.e., energy span model). Did the authors consider plotting the same potential energy surface under an applied potential (i.e., see Nørskov and co-workers) to ensure that the reaction is spontaneous?
- vii) Considering the high N₂ concentration in flue gas, how does the catalyst work in the presence of N₂ (only) and under a CO₂ + N₂ feeding environment?

3) Revisions

- i) The pore size in-text is mentioned in angstroms but in nm in Figure 1, Line 146. The authors should consider using similar units throughout the manuscript.
- ii) The N₂ sorption diagrams for the important species can be plotted together to effectively visualize the enhancements in the close-DAE-COFs.
- iii) The SEM and TEM images (Fig. 2d, 7a-d) are hard to compare with different scales for different species. Would it be possible to employ similar scales?
- iv) Fig. 4 c-d does not provide any indication regarding the green and grey lines.
- v) XPS: The relevant information or references that associate the spectral signatures with the respective Co-orbitals should be included.
- vi) The authors should try to relate the experimental increase in efficiency (1.2-fold increase) to the difference in free energy for the rate-determining step (0.04 eV). The computational energies point towards five times increase in the reaction rate, which agrees with the experimental observations.
- vii) The DOS plots must point out the Fermi level. There is no discussion on why there are additional peaks above and below the fermi level in the case of the close-DAE-COF, which are not present in open-DAE-COF. This information may be relevant for further catalyst design.

Reviewer #4 (Remarks to the Author):

H.-J. Zhu, et al., reported a reticular chemistry-based electrocatalysts for efficient CO₂ reduction reaction (CO₂RR) under aerobic environments by photoswitching derived oxygen passivation strategies. Direct CO₂RR with flue gas would receive great attention as a practical carbon capture utilization (CCU) approach. Idea of applying photonicswitching unit 1,2-Bis(5'-formyl-2'-methylthien-3'-yl)cyclopentene (DAE) to covalent organic framework (COF) is innovative. Comprehensive studies including material synthesis, characterization, in-situ spectroscopy, CO₂RR performance measurement, and theoretical computation are impressive and supports their conclusion. For the better clarity, this reviewer has several concerns about open/close modes of DAE, which needs to be studied more in detail. I recommend major revision of this manuscript for the publication in Nature Communications. Details as follows:

1. It was explained that 3 h exposure of open-DAE-BPy-CoPor to UV light is required to derive photocyclization and close-DAE-BPy-CoPor. Why does the photoswitching require 3 h for the transition of DAE? If the exposure time becomes shorter or longer, what happens to the openness of DAE-BPy-CoPor? How can we understand the kinetics of photoswitching?
2. How can we distinguish the degree of openness of DAE-BPy-CoPor? Although authors provided XPS S 2p (Fig. 4d) and evaluation of O₂-toleration, it is difficult to find a methodology to determine whether all DAEs are fully closed or not.

3. In the CO₂RR performance results, authors show product distribution of CO in 100% CO₂ and aerobic conditions. I recommend to providing all Faradaic efficiencies (FEs) for CO₂RR and HER such as CO and H₂ FE together.

4. In the 100% CO₂ condition, except for the potential of -0.5 V (vs. RHE), three samples (Close/Open-DAE-BPy-CoPor, and BPy-CoPor) exhibit similar CO selectivities. Why does BPy-CoPor show degraded CO selectivity compared to others especially at -0.5 V (vs. RHE)?

5. In the CO₂RR under aerobic conditions, I understand the superior CO₂RR performance of the close-BPy-CoPor in Fig. 5e. Although open-BPy-CoPor shows lower CO FE compared to close-BPy-CoPor, BPy-CoPor has a huge drop in CO FE more than open-BPy-CoPor. What is the reason for this phenomenon? Based on operando ATR-FTIR, open-BPy-CoPor also exhibits *OOH under CO₂+O₂ mixed gas condition. I recommend the measurement of operando ATR-FTIR of Bpy-CoPor under CO₂+O₂ mixed gas to discover the difference between open-BPy-CoPor and Bpy-CoPor.

6. Authors mentioned that there was no Co or CoO_x based nanoparticle formation after CO₂RR with Supplementary Figs. 22-23, which show TEM images. I recommend investigating XRD and FT-IR to confirm the stability of COF after CO₂RR.

7. In Supplementary Figure 16a, there is a typo; both black and red denote after states. Furthermore, when we check the difference between before/after of DPBF absorption spectrum, close- and open-DAE BPy-CoPor exhibit 0.093 and 0.11. Are they different enough to see the difference between them?

Reviewer #1 (Remarks to the Author):

1. The authors reported a cobalt porphyrin-based covalent organic framework integrated with 1,2-Bis(5'-formyl-2'-methylthien-3'-yl)cyclopentene (DAE) moiety (DAE-BPy-CoPor), of which the framework shows ring-closing/opening features for modulating the oxygen-passivation behaviors during CO₂RR. However, the characterization and mechanism of the oxygen passivation effect is very insufficient, and the conclusion is too simplistic and crude.

Response: Thanks for your comment. According to the Reviewer's suggestions, we added pertinent tests and integrated them with the characterization and mechanism of the oxygen passivation effect study from the original manuscript to prove our propose, including UV visible absorption, ORR test, operando electrochemical attenuated total reflection Fourier transform infrared spectroscopy (ATR-FTIR) experiments and the projected electron density (PDOS) calculation. We used 1,3-diphenylisobenzofuran (DPBF) that sensitive to oxygen species to detect the reactive oxygen species for COFs. In the presence of oxygen species, DPBF can immediately form an unstable endoperoxide which decomposes to 1,2-dibenzoylbenzene. DPBF exhibits a strong absorption band at 415 nm, and the decrease in adsorption at this wavelength proportional to the amount of the oxygen species generated. As shown in Figures R1 and R2, the close-DAE-BPy-CoPor has the weakest degradation, indicating excellent O₂ toleration ability of close-DAE-BPy-CoPor. We also examined the ORR performance for COFs (Figure R3), which showed that the close-DAE-BPy-CoPor exhibited the lowest oxygen activity with the most negative half-wave potential and the smallest diffusion-limiting current density compared with open-DAE-BPy-CoPor and BPy-CoPor. Furthermore, the operando electrochemical attenuated total reflection Fourier transform infrared spectroscopy (ATR-FTIR) experiments of COFs were conducted to probe the deep mechanism of oxygen passivation during CO₂RR (Figures R4 and R5). The key intermediate (*OOH) of ORR for the open-DAE-BPy-CoPor was detected, but there was no obvious band in the close-DAE-BPy-CoPor. Besides, we also combined the projected electron density (PDOS) to reveal the oxygen passivation mechanism of close-DAE-BPy-CoPor

(Figure R6), which indicated that the O₂ in the open-DAE-BPy-CoPor was closer to the Fermi level than close-DAE-BPy-CoPor catalyst and exhibited the weaker hybridization between O₂ and Co active sites in the close-DAE-BPy-CoPor, which was well agreed that the electrons on close-DAE-BPy-CoPor were hardly transfer to O₂, thus passivating O₂ on close-DAE-BPy-CoPor under aerobic condition.

We have added the Figure R2 and Figure R5 in the revised Supplementary Information (Page 22, Supplementary Figure 20; page 43, Supplementary Figure 41) and related description as follows in the revised manuscript (Page 13, lines 297 to 302; page 17, lines 417 to 426).

“As shown in Supplementary Figs. 19 and 20, the absorbance at $\lambda = 410$ nm showed the largest degradation in the BPy-CoPor (0.16, 19% of DPBF absorption decreased), meanwhile, the close-DAE-BPy-CoPor displayed the weakest degradation (0.093, 12% of DPBF absorption decreased), indicating excellent O₂ toleration ability of close-DAE-BPy-CoPor.”

“Furthermore, the operando ATR-FTIR measurement of BPy-CoPor and open-DAE-BPy-CoPor under CO₂ + O₂ mixed gas showed that the band located at 1398 cm⁻¹ in the open-DAE-BPy-CoPor and BPy-CoPor spectra under co-feeding CO₂ and O₂ was assigned to a carboxyl intermediate of *COOH (Supplementary Fig. 41). More importantly, in the BPy-CoPor curve, an obviously downward broad band at 1637 cm⁻¹ was observed, which was assigned to the H-O-H bending generation. It suggested a hint of water production during the reaction. Furthermore, the intensity of H-O-H bending band is stronger than *COOH band, which indicated the ORR rather than CO₂RR was more easily happened for BPy-CoPor.”

Figure R1. The DPBF absorption spectrum of **a** BPy-CoPor, **b** open-DAE-BPy-CoPor and **c** close-DAE-BPy-CoPor in 0.1 M TBAPF₆/MeCN at -0.1 V vs. Ag/AgCl.

Figure R2. The decay rate of DPBF upon close-DAE-BPy-CoPor, open-DAE-BPy-CoPor and BPy-CoPor.

Figure R3. Linear sweep voltammograms (LSVs) of BPy-CoPor, open-DAE-BPy-CoPor and close-DAE-BPy-CoPor at 1600 rpm in O₂-saturated 0.2 M Na₂SO₄ (scan rate: 10 mV s⁻¹).

Figure R4. Operando ATR-FTIR spectra on close-DAE-BPy-CoPor and in open-DAE-BPy-CoPor in the CO₂ saturated 0.5 M KHCO₃ and co-feeding CO₂ and 5% O₂.

Figure R5. a Operando ATR-FTIR spectra on BPy-CoPor in the co-feeding CO₂ and 5% O₂ 0.5 M KHCO₃. **b** The comparison operando ATR-FTIR spectra of close-DAE-BPy-CoPor, open-DAE-BPy-CoPor and BPy-CoPor in the co-feeding CO₂ and 5% O₂ 0.5 M KHCO₃.

Figure R6. Projected density of states and integrated density of states of adsorption structures of O₂ on close-DAE-BPy-CoPor and open-DAE-BPy-CoPor.

2. In both CO₂-saturated and CO₂/O₂ co-feeding condition, the close-DAE-BPy-CoPor showed a moderate higher CO partial current density than that of open-DAE-BPy-CoPor, which should be mainly responsible to their different electronic conductivity of the frameworks. Therefore, it is too preliminary to conclude that the slight activity enhancement of close-DAE-BPy-CoPor results from the oxygen-passivation, instead of its higher electronic conductivity.

Response: We highly appreciate the suggestions to improve the quality of our paper. As stated by the reviewer, electronic conductivity of the frameworks can significantly affect the total current density, which has been revealed by our previously published paper (*ACS Energy Lett.* 2020, 5, 1005). To confirm whether the electronic conductivity of the frameworks affect the CO partial current density (j_{CO}) over close-DAE-BPy-CoPor in this system, we tested the electrical conductivities of close-DAE-BPy-CoPor, open-DAE-BPy-CoPor and BPy-CoPor. As shown in Figure R7, close-DAE-BPy-CoPor ($2.55 \times 10^{-8} \text{ S m}^{-1}$) has a higher electrical conductivity than those of open-DAE-BPy-CoPor ($3.40 \times 10^{-9} \text{ S m}^{-1}$) and BPy-CoPor ($6.60 \times 10^{-9} \text{ S m}^{-1}$). Therefore, close-DAE-BPy-CoPor exhibited large total current densities than

that the other two materials at the applied potentials from -0.5 to -0.9 V vs RHE (Figure R8a). So, the electronic conductivity of the frameworks indeed significantly affects the total current density and the higher electron conductivity could facilitate in the electron transfer to the active sites during CO₂RR, thus the close-DAE-BPy-CoPor exhibited higher total current density (Figure R8a). However, it should be noted that the CO partial current density (j_{CO}) not only affected by the total current density, but also associated with the CO selectivity. In pure CO₂-saturated condition, the three materials show similar CO Faradic efficiency (FE_{CO}) at -0.6 to -0.9 V (Figure R8b), thus, the close-DAE-BPy-CoPor shows the highest j_{CO} because it has the best electron conductivity and total current density (Figure R8c). While in the CO₂/O₂ co-feeding electrolyte, close-DAE-BPy-CoPor shows obviously larger CO/FE_{CO} than open-DAE-BPy-CoPor and BPy-CoPor (Figure R9). So, close-DAE-BPy-CoPor exhibited highest j_{CO} , although BPy-CoPor has the best total current density values (Figure R10). It suggested that the CO selectivity significantly affects the j_{CO} . As well know, the CO selectivity is associated with the real active site. In our system, the density functional theory (DFT) calculations (Figure R11) revealed the close-DAE-BPy-CoPor could make the O₂ passivation and increase the barrier to form the *OOH intermediate, thereby suppress the ORR reaction under CO₂/O₂ co-feeding. Therefore, compared with the open-DAE-BPy-CoPor and BPy-CoPor, the CO₂RR was occurred preferentially than the ORR over close-DAE-BPy-CoPor, which shows higher CO selectivity.

We have added the Figure R7 in the revised Supplementary Information (Page 19, Supplementary Figure 17) and related description as follows in the revised manuscript (Page 12, lines 279 to 282).

“As expected, the electrical conductivity of close-DAE-BPy-CoPor has an order of magnitude improvement ($2.55 \times 10^{-8} \text{ S m}^{-1}$) than open-DAE-BPy-CoPor ($3.40 \times 10^{-9} \text{ S m}^{-1}$) and BPy-CoPor ($6.60 \times 10^{-9} \text{ S m}^{-1}$) (Supplementary Fig. 17).”

Figure R7. Electrical measurement of **a** close-DAE-BPy-CoPor ($L = 0.51$ mm), **b** open-DAE-BPy-CoPor ($L = 0.25$ mm) and **c** BPy-CoPor ($L = 0.48$ mm) were performed using two-electrode in air at a constant temperature of 298 K and absence of light ($\sigma = L/(R \times \pi(d/2)^2)$, $d = 2.5$ mm).

Figure R8. The electrocatalytic CO_2RR performances. **a** Linear sweep voltammetry curves, **b** the CO Faradic efficiency and **c** the CO partial current density of the BPy-CoPor, close-DAE-BPy-CoPor and open-DAE-BPy-CoPor.

Figure R9. The CO partial current density of the BPy-CoPor, close-DAE-BPy-CoPor and open-DAE-BPy-CoPor in the co-feeding CO_2 and 5% O_2 .

Figure R10. The LSV of close-DAE-BPy-CoPor, open-DAE-BPy-CoPor and BPy-CoPor in the CO₂RR with co-feeding CO₂ and O₂.

Figure R11. Density functional theory calculations. **a** Projected density of states and integrated density of states of adsorption structures of O₂ on close-DAE-BPy-CoPor and open-DAE-BPy-CoPor. **b** The free energy of ORR pathways for close-DAE-BPy-CoPor and open-DAE-BPy-CoPor.

3. It is too strange that the CO partial current densities of the catalysts conducted in a CO₂-saturated condition are higher than those gained in the co-feeding CO₂ and 5% O₂, even the competitive ORR occurs.

Response: We think the reviewer's question may be that the CO partial current densities of the catalysts conducted in a CO₂-saturated condition are lower than those gained in the co-feeding CO₂ and 5% O₂. Not only you felt strange about the phenomenon, the corresponding author (Yuan-Biao Huang) also felt surprise when the first author obtained the data and I asked her to repeat this measurement and the same

result was obtained. It indeed is very difficult to understand this phenomenon. In order to understand this question, we carefully summarized and compared the performance of the CO selectivity and total current density of close-DAE-BPy-CoPor under pure CO₂ conditions and co-feeding CO₂ + O₂ (Figure R12). As well know, the total current density (I_{total}) is contributed by CO₂RR and HER in pure CO₂-saturated condition, while the total current density is contributed by CO₂RR, HER and ORR under co-feeding CO₂ + O₂ electrolyte. As shown in Figure R12a, the total current density of close-DAE-BPy-CoPor under co-feeding CO₂ + O₂ is higher than pure CO₂-saturated electrolyte. But, the FE_{CO} of close-DAE-BPy-CoPor significantly decreased from -0.5 to -0.9 V (Figure R12b). It is worth noting that the CO partial current density (j_{CO}) is related to the total current density and the FE_{CO} ($j_{\text{CO}} = I_{\text{total}} \times \text{FE}_{\text{CO}}$). As shown in Figure R12a and b, due to the decreased FE_{CO} rate is higher than the I_{total} increased rate at -0.5 V (I_{total} increase rate, 5.8; FE_{CO} decrease rate, 10.9), the j_{CO} of close-DAE-BPy-CoPor in CO₂ + O₂ are lower than those in pure CO₂-saturated electrolyte at -0.5 to -0.9 V (Figure R12c). When at -0.6 V and -0.7 V, it shows the similar increased I_{total} rate and the decreased FE_{CO} rate, thus close-DAE-BPy-CoPor has similar j_{CO} value under pure CO₂ or co-feeding CO₂ + O₂ condition. While at -0.8 V and -0.9 V, the increased I_{total} rate is higher than the decreased FE_{CO} rate, close-DAE-BPy-CoPor shows an increased j_{CO} value at co-feeding CO₂ + O₂ condition than in pure CO₂ electrolyte. Interestingly, with further increasing the applied potential, the CO₂RR play a dominant role, and the FE_{CO} and I_{total} increase significantly at co-feeding CO₂ + O₂ condition. So, the j_{CO} increased significantly under co-feeding CO₂ + O₂ than pure CO₂ condition at -1.0 and -1.1 V. As shown in Figure R12, the CO partial current densities of the catalysts conducted in a CO₂-saturated condition and co-feeding CO₂ and 5% O₂ are closely related to the applied potentials. The phenomenon that the CO partial current density in the co-feeding CO₂ and O₂ was higher than CO₂-saturated condition at high applied potentials is also found other catalyst system, such as the Cu catalyst (Qi Lu et al. *Nat. Commun.*, 2020, 11, 3844).

We have added the Figure R12 in the revised Supplementary Information (Page 36,

Supplementary Figure 34) and related description as follows in the revised manuscript (Page 16, lines 392-395).

“Besides, the j_{CO} of close-DAE-BPy-CoPor can reach up to -20.1 mA cm^{-2} at -1.0 V , which was higher than BPy-CoPor ($j_{CO} = -7.36 \text{ mA cm}^{-2}$) and open-DAE-BPy-CoPor ($j_{CO} = -15.4 \text{ mA cm}^{-2}$) at the same potential (Supplementary Figs. 33 and 34, Fig. 5f).”

Figure R12. The CO₂RR performance of close-DAE-BPy-CoPor under pure CO₂ and co-feeding CO₂ + O₂ condition. **a** Total current density. **b** FE_{CO}. **c** j_{CO} . The numbers stand for the enhancement relative to the rates at pure CO₂.

4. Actually, the catalytic activity of close-DAE-BPy-CoPor is quite poor, also inferior to most of recently reported catalysts. For example, in CO₂-saturated CO₂RR, close-DAE-BPy-CoPor only showed a low partial current density of -8.5 mA cm^{-2} at -0.9 V . Moreover, the stability is low, as both current density and Faradaic efficiency dropped quickly during the stability test.

Response: Thanks for your constructive comment. It should be noted that the current density under the H-type cell is strongly limited by the poor CO₂ solubility. We summarized the reported cobalt porphyrin-based electrocatalysts in H-type cell with similar test conditions as close-DAE-BPy-CoPor. As shown in Table R1, the CO partial current density of close-DAE-BPy-CoPor in CO₂-saturated electrolyte shows -8.5 mA cm^{-2} at -0.9 V which is higher than most of reported cobalt porphyrin-based electrocatalysts including BPy-CoPor, ViB12@rGO, Co-TTCOF and COF-366-Co. Besides, we also have taken various measures to further enhance the CO partial

current density, such as updating the electrolytic cell device (gas diffusion electrode cell (GDE)) and changing the electrolyte (0.1 M KOH and 1 M KOH). As shown in Figure R13a, the FE_{CO} of close-DAE-BPy-CoPor in the GDE system can reach $\geq 90\%$ in the wide potential range from -0.6 to -1.1 V in 0.5 M $KHCO_3$. Besides, the CO partial current density (j_{CO}) reached -15.0 mA cm^{-2} at -0.9 V, which was almost 1.8-fold higher than close-DAE-BPy-CoPor in the H-cell (Figure R13b). Electrolytes have been considered to highly participate in the CO_2RR process via their interactions with catalyst surface, reactants, intermediates, and even the products. Thus, different electrolytes will also greatly effect on the current density of electrocatalyst. We further study the CO_2RR performance of close-DAE-BPy-CoPor in 0.1 M KOH and 1 M KOH. As shown in Figure R13c, the FE_{CO} of close-DAE-BPy-CoPor was above 90% in a wide potential range (-0.3 V to -1.3 V), and the partial current density of CO can reach -44.2 mA cm^{-2} at -1.3 V in 0.1 M KOH, which is superior to most reported porphyrin-based catalysts. Besides, the FE_{CO} of close-DAE-BPy-CoPor was above 90% in the potential range from -0.3 to -0.5 V, and the CO partial current density can reach $-108.9 \text{ mA cm}^{-2}$ at an applied potential of -0.8 V in 1 M KOH (Figure R13d).

As for the stability, we compared with some porphyrin-based catalysts under similar CO_2RR conditions (Table R1), such as TTF-Por(Co)-COF, Co-TPP-cov, Co-TOO/CNT and Co-Bpy-COF-Ru1/2, the stability of close-DAE-BPy-CoPor is relatively excellent. But your suggestion is very useful to improve the quality of our manuscript, thus we have fine-tuned the catalytic potential for close-DAE-BPy-CoPor, enabling a stable current density for 24 h at -0.9 V (vs. RHE), while maintaining a CO selectivity exceeding 80% (Figure R14).

We have added Table R1 and Figures R13-R14 in the revised Supplementary Information (Page 57, Supplementary Table 4, page 27; Supplementary Figure 25, and page 30, Supplementary Figure 28) and the related description as follows in the revised manuscript (Page 14, lines 326-330, page 14, lines 337-248 and pages 15-16, lines 367-373).

“As shown in Fig. 5b, the close-DAE-BPy-CoPor exhibited remarkable Faradaic

efficiencies of CO (FE_{CO}) ($\geq 90\%$) across the entire potential window from at -0.6 V to -0.9 V (vs. RHE), which were larger than those of BPy-CoPor and open-DAE-BPy-CoPor under pure CO_2 gas (Supplementary Fig. 23 and Supplementary Table 4)."

"Besides, we conducted an in-depth investigation into the performance of close-DAE-BPy-CoPor in a gas diffusion electrode cell (GDE). As shown in Supplementary Fig. 25, the CO partial current density (j_{CO}) reached -15.0 mA cm^{-2} at -0.9 V in a CO_2 -saturated condition. Additionally, we further study the CO_2RR performance of close-DAE-BPy-CoPor in 0.1 M KOH and 1 M KOH. As illustrated in Supplementary Figs. 25c and d, the FE_{CO} of close-DAE-BPy-CoPor was above 90% in a wide potential range (-0.3 V to -1.3 V), and the partial current density of CO can reach -44.2 mA cm^{-2} at -1.3 V in 0.1 M KOH, which was superior to most reported porphyrin-based catalysts. Furthermore, the FE_{CO} of close-DAE-BPy-CoPor was above 90% in the potential range from -0.3 to -0.5 V, and the CO partial current density can reach -108.9 mA cm^{-2} at an applied potential of -0.8 V in 1 M KOH."

"The long-term stability of close-DAE-BPy-CoPor was studied by chronoamperometric test in a CO_2 -saturated 0.5 M $KHCO_3$ electrolyte, which showed the corresponding FE_{CO} can be retained ($FE_{CO} \geq 80\%$) and there was 36.2 % current drop (-5.20 mA cm^{-2} drop to -3.32 mA cm^{-2}) at -0.7 V after 24 h (Fig. 5d). We have fine-tuned the catalytic potential for close-DAE-BPy-CoPor, enabling a stable current density for 24 h at -0.9 V (vs. RHE), while maintaining a CO selectivity exceeding 80% (Supplementary Fig. 28)."

Table R1. The summary of CO_2 electroreduction performances for reported electrocatalysts and this work.

Catalyst	electrolyte	Highest FE_{CO} (%)	j_{CO} (mA cm^{-2})	Stability (h)	Ref.
Close-DAE-BPy-CoPor	0.5 M $KHCO_3$	98.0	-8.47 (-0.9 V)	24	This work
Open-DAE-BPy-CoPor	0.5 M $KHCO_3$	95.2	-6.99 (-0.9 V)	NA	This work
BPy-CoPor	0.5 M	94.1	-5.30	NA	This work

ViB ₁₂ @rGO	KHCO ₃ 0.5 M KHCO ₃	94.5	(-0.9 V) -6.24 (-0.8 V)	10	ACS Appl Mater Interfaces 12, 41288-41293 (2020)
Co-TTCOF	0.5 M KHCO ₃	91.3	-1.84 (-0.7 V)	40	Nat Commun 11, 497 (2020)
COF-366-Co	0.5 M KHCO ₃	90.0	-1.8 (-1.1 V)	24	Science 349, 1208 (2015)
COF-367-Co	0.5 M KHCO ₃	91.0	-3.3 (-1.1 V)	24	Science 349, 1208 (2015)
TTF-Por(Co)-COF	0.5 M KHCO ₃	70.0	-6.88 (-0.9 V)	10	ACS Energy Letters 5, 1005-1012 (2020)
Co-TPP-cov	0.5 M KHCO ₃	67	-1.065 (-0.63 V)	4	Angew. Chem. Int. Ed. 56, 6468-6472 (2017)
Co-TPP/CNT	0.5 M KHCO ₃	91	-3.2 (-0.66 V)	12	Angew. Chem. Int. Ed. 58, 6595-6599 (2019)
Co-Bpy-COF-Ru1/2	0.5 M KHCO ₃	96.7	about -9 (-0.7 V)	13	J Am Chem Soc , (2023)

Figure R13. The electrocatalytic CO₂RR performances of close-DAE-BPy-CoPor. **a** The FE_{CO} and **b** the j_{CO} in 0.5 M KHCO₃ using GDE and H-cell. **c** The FE_{CO} and **d** the j_{CO} in 0.5 M KHCO₃, 0.1 M KOH and 1 M KOH using GDE.

Figure R14. Stability of close-DAE-BPy-CoPor at -0.9 V vs. RHE under CO₂-saturated 0.5 M KHCO₃ aqueous solution.

5. The ring-closing/opening forms of DAE-BPy-CoPor could lead to the electronic regulation on the CoN₄ centers and thus their different catalytic activity. To identify the electronic difference, the Co K-edge of EXAFS spectra both open-DAE-BPy-CoPor and closing-DAE-BPy-CoPor should be provided.

Response: As per your suggestion, we tried our best to test the X-ray absorption

spectra of close-DAE-BPy-CoPor to illustrate the electronic structure on the CoN_4 centers. As shown in Figure R15a, the Co K-edge X-ray absorption near-edge (XANES) profile of close-DAE-BPy-CoPor exhibit a similar wave feature with that of open-DAE-BPy-CoPor, indicating that the Co center valence of close-DAE-BPy-CoPor similar with open-DAE-BPy-CoPor. The Fourier-transformed of Co K-edge extended X-ray absorption fine structure (EXAFS) spectra present a dominate peak at 1.45 Å, which could be attributed to the Co-N bond. Besides, compared with open-DAE-BPy-CoPor, the Co-N peak of close-DAE-BPy-CoPor is negatively shifted ($\Delta = 0.02$ Å), indicating a reduced bond length of Co-N in close-DAE-BPy-CoPor (Figure R15b). The EXAFS fitting results further prove that the bond length of Co-N in close-DAE-BPy-CoPor (1.93 Å) is shorter than that of Co-N in open-DAE-BPy-CoPor (1.95 Å) (Table R2). The contraction of the Co-N bond length can facilitate transfer electrons to the Co center, indicating that the Co center electrons of close-DAE-BPy-CoPor is more than open-DAE-BPy-CoPor. The phenomenon that the alterations in the catalytic center electrons are reflected in the bond lengths explored in the EXAFS is also found in other catalyst system, such as the Hg-CoTPP (Wai-Yeung Wong et al. *J. Am. Chem. Soc.* 2022, 144, 15143-15154) and NiPc-CN molecularly dispersed electrocatalysts (Yongye Liang et al. *Nat. Energy.* 2020, 5, 684-692).

We have added the Figure R15 and Table R2 in the revised Supplementary Information (Page 17, Supplementary Figure 15; page 56, Supplementary Table 3) and related description (Pages 11-12, lines 258-273) and references (Page 28-29, lines 734-740) as follows in the revised manuscript.

“Furthermore, the X-ray absorption spectra of close-DAE-BPy-CoPor were collected to illustrate the electronic structure on the CoN_4 centers. The Co K-edge X-ray absorption near-edge (XANES) profile of close-DAE-BPy-CoPor exhibited a similar wave feature with that of open-DAE-BPy-CoPor, indicating that the Co center valence of close-DAE-BPy-CoPor similar with open-DAE-BPy-CoPor. The Fourier-transformed of Co K-edge extended X-ray absorption fine structure (EXAFS) spectra presented a dominate peak at 1.45 Å, which could be attributed to the Co-N

bond. Besides, compared with open-DAE-BPy-CoPor, the Co-N peak of close-DAE-BPy-CoPor was negatively shifted ($\Delta = 0.02 \text{ \AA}$), indicating a reduced bond length of Co-N in close-DAE-BPy-CoPor (Supplementary Fig. 15). The EXAFS fitting results further proved that the bond length of Co-N in close-DAE-BPy-CoPor (1.93 \AA) was shorter than that of Co-N in open-DAE-BPy-CoPor (1.95 \AA). The contraction of the Co-N bond length can facilitate transfer electrons to the Co center^{56,57}, indicating that the Co center electrons of close-DAE-BPy-CoPor was more than open-DAE-BPy-CoPor (Supplementary Table 3).”

“56. Fang, M., Xu, L., Zhang, H., Zhu, Y., Wong, W.-Y. Metalloporphyrin-linked mercurated graphynes for ultrastable CO₂ electroreduction to CO with nearly 100% selectivity at a current density of 1.2 A cm⁻². J. Am. Chem. Soc. **144**, 15143-15154 (2022).

57. Zhang, X., et al. Molecular engineering of dispersed nickel phthalocyanines on carbon nanotubes for selective CO₂ reduction. Nat. Energy **5**, 684-692 (2020).”

Figure R15. The local coordination structure. a Co K-edge of X-ray absorption near-edge structure spectra of open-DAE-BPy-CoPor and close-DAE-BPy-CoPor. **b** Co K-edge of EXAFS spectra of open-DAE-BPy-CoPor and close-DAE-BPy-CoPor. **c** The extended X-ray absorption fine structure fitting curves of close-DAE-BPy-CoPor.

Table R2. Fitting results from EXAFS analysis of open-DAE-BPy-CoPor. (CN: coordination number; R: distance between absorber and backscatter atoms; σ^2 : Debye-Waller factor (a measure of thermal and static disorder in absorber-scatterer

distances); ΔE_0 : the inner potential correction; R factor is used to value the goodness of the fitting.)

Sample	Path	CN	R(Å)	$\sigma^2(10^{-3} \text{ \AA}^2)$	ΔE_0 (eV)	R factor
Open-DAE-BPy-CoPor	Co-N	4.3 ± 0.3	1.95	6.32	-6.48	0.02
Close-DAE-Bpy-CoPor	Co-N	4.0 ± 0.6	1.93	4.97	-1.21	0.003

Reviewer #2 (Remarks to the Author):

Response: Thank you very much for your comments. We have tried our best to revise our manuscript, and responded to all reviewers' questions one by one.

Reviewer #3 (Remarks to the Author):

1) Key Results

In this study, Cao, Huang, and co-workers report using a photoswitching unit (DAE) integrated into a cobalt porphyrin-based covalent organic framework (COF) for the CO₂ reduction reaction (CO₂RR) under an aerobic environment. The authors have identified a significant problem in the CO₂RR reduction strategy, suppressing the oxygen evolution reaction (ORR) while selectively reducing CO₂ to value-added products. The authors have provided adequate experimental evidence for synthesizing the COFs, including spectroscopic techniques such as SEM, TEM, and PXRD. They have also employed XPS to provide local coordination structure of their materials, a significant part of the work, and performed DFT calculations to probe the reaction mechanism.

The electrochemical performance evaluation also constitutes a noteworthy part of the work, especially where all critical variations of the COFs are considered, and

proper trends are reflected and discussed accordingly. It is important to note that the close-DAE-COF obtained is stable over 24 hours and can maintain a good Faradaic efficiency (FE) for CO₂RR.

Overall, Cao, Huang, and co-workers have successfully achieved their strategy to make an O₂-tolerant COF for CO₂RR where the efficiency can be controlled by light.

Response: Thank you very much for your supporting and positive comments.

2) Conclusions, claims & additional evidence.

The experimental work presented here does support the mentioned conclusions. However, there are a few methodologies and instrumentation information that are missing, critical for the reproducibility of the results:

i) Isotopic label experiments. Were the isotopic species procured from commercial sources or synthesized in the lab?

Response: In this work, the isotopic species was purchased from WUHAN NEWWRADAR SPECIAL GAS Co., LTD and the enrichment is 99 atom% of ¹³C.

We have now included relative information as follows in the revised manuscript within the part of characterizations and instruments (Page 22, lines 543-544).

“The isotopic species was purchased from WUHAN NEWWRADAR SPECIAL GAS Co., LTD and the enrichment is 99 atom% of ¹³C”

ii) GC & ICP-MS – The methodology and instrument details are missing in the “Characterizations and Instruments” section.

Response: Thanks for your kind suggestion. According to your suggestion, we have added the details methodology and instrument information for GC & ICP-MS in “Characterizations and Instruments” section as follows in the revised manuscript (Page 22, lines 539-543).

“The gas chromatography measurements were performed on the FULLI INSTRUMENTS GC9790 PLUS gas chromatograph (GC) equipped with FID and TCD. The analysis of metal content was measured by inductively coupled plasma

atomic emission spectroscopy on an Avio220Max.

The computational part is also missing information central to the reproduction of results, such as:

iii) The Bader charge on the Co-atoms was counted, but the methodology implemented here is not mentioned; for example, how were the Co-atoms counted in the analysis? (Lines 406-408)

Response: Thanks for your suggestions. The Bader charge was calculated by density functional theory through the VASP package. First, the intermediate structures were optimization, then the single Energies of the optimization intermediates were performed with the key words LAECHG=.TRUE.. Herein, the nuclear charge was written to AECCAR0 and the valence charge was written to AECCAR2. The two charge density files can be summed using the chgsum.pl script. The following output files are generated: ACF.dat, BCF.dat and AtomVolumes.dat.

ACF.dat contains the coordinates of each atom, the charge associated with it according to Bader partitioning, percentage of the whole according to Bader partitioning and the minimum distance to the surface.

BCF.dat contains the coordinates of each Bader maxima, the charge within that volume, the nearest atom and the distance to that atom.

AtomVolumes.dat contains the number of each volume that has been assigned to each atom.

We can get the Bader charge by total charge minus valence electrons, which obtained from ACF.dat and POTCAR, respectively.

Reference

<http://theory.cm.utexas.edu/henkelman/code/bader/>

iv) The computed Cartesian coordinates should be provided in the supporting information.

Response: As per your suggestion, we have added the computed Cartesian

coordinates of close-DAE-BPy-CoPor and open-DAE-BPy-CoPor for CO₂RR, HER and ORR in the Supplementary Information for calculation (Supplementary Tables 5-26).

We have added the related description as follows in the revised manuscript (Page 18, lines 440-442).

“The calculation intermediate models of the CO₂RR, HER and ORR were presented in the supplementary information from Supplementary Fig. 42 to Supplementary Fig. 47, and Supplementary Table Supplementary Tables 5-26.”

v) The authors should consider discussing how the density of states calculations (projected and integrated) were obtained.

Response: Thanks for your suggestion. The density of states (DOS) was calculated by density functional theory using the VASP package. First, the intermediate structures were optimized, then the single energies of the optimized intermediates were performed with the high accurate, and the CHGCAR file was generated. The DOS of intermediates were further calculated by the key words NEDOS= 2000~3000, ISMEAR = -5, LORBIT = 10 or 11, ICHGCAR = 1 or 11.

The related Settings are explained as follows:

The larger the NEDOS value is set, the more accurate the DOS interval will be distinguished. In general, NEDOS = 2000 is accurate enough.

For the calculation of the total energy in bulk materials, we recommend the tetrahedron method with Blöchl corrections (ISMEAR=-5). This method also gives a smoothly nice electronic DOS.

LORBIT = 10 means that the DOS is decomposed into each atom and the s, p, d orbital of the atom, which called the Local density of states (LDOS). LORBIT = 11 indicates that the DOS is further decomposed into p_x, p_y, p_z and other orbitals, called Projected state density (PDOS) or Partial state density (PDOS).

The calculated results of DOS were analyzed by the VASPKIT package, the projected density of states and integrated projected density of states can be summed by the “Option 11”.

<https://cms.mpi.univie.ac.at/wiki/index.php/NEDOS>

<https://cms.mpi.univie.ac.at/wiki/index.php/ISM EAR>

<https://www.vasp.at/wiki/index.php/Category:Examples>

<https://vaspkit.com/tutorials.html#density-of-states>

vi) The overall free energy surface mentioned is endothermic, indicating that the reaction would not occur (i.e., energy span model). Did the authors consider plotting the same potential energy surface under an applied potential (i.e., see Nørskov and co-workers) to ensure that the reaction is spontaneous?

Response: Thanks for your useful suggestion. The free energy in previous study was calculated at $U = 0$ V (U is the electrode potential), which exhibited the free energy surface mentioned is endothermic. In addition, we calculated the free energies of CO₂RR at $U = -0.7$ V (J. K. Nørskov et. al, *J. Phys. Chem. Lett.* 2015, 6, 2663; J. K. Nørskov et. al, *J. Phys. Chem. Lett.* 2016, 7, 1686), and the overall free energy surface mentioned is exothermic (Figure R16). As shown in Fig. 5b, the close-DAE-BPy-CoPor also showed excellent CO₂RR performance with achieved almost 100% FE_{CO} at -0.7 V.

We have added the Figure R16 in the revised Supplementary Information (Page 50, Supplementary Figure 48), and related description (Page 19, lines 453-455) and references (Page 29, lines 763-766) as follows in the revised manuscript.

“Besides, we recalculated the free energies of CO₂RR at $U = -0.7$ V, and the overall free energy surface mentioned was exothermic^{65,66}(Supplementary Fig. 48).”

*“65. Chan, K., Nørskov, J. K. Electrochemical barriers made simple. *J. Phys. Chem. C.* 6, 2663-2668 (2015).*

*66. Chan, K., Nørskov, J. K. Potential dependence of electrochemical barriers from ab initio calculations. *J. Phys. Chem. C.* 7, 1686-1690 (2016).”*

Fig. 5b The CO Faradic efficiency of the BPy-CoPor, close-DAE-BPy-CoPor and open-DAE-BPy-CoPor.

Figure R16. Energy diagrams CO₂RR and HER of close-DAE-BPy-CoPor and open-DAE-BPy-CoPor vs the electrode potential at -0.7 V in H-cell.

“The effect of a bias on all states involving an electron in the electrode was considered, by shifting the energy of this state by $\Delta G_U = -eU$, where U is the electrode potential.”

vii) Considering the high N₂ concentration in flue gas, how does the catalyst work in

the presence of N₂ (only) and under a CO₂ + N₂ feeding environment?

Response: Thanks for your useful suggestion. We added the CO₂RR performance test of close-DAE-BPy-CoPor under the presence of N₂ (only) and under a CO₂ + N₂ feeding environment. As shown in Figure R17, only H₂ was obtained under only N₂ feeding environment, which suggested that the obtained CO was originated from CO₂. Under the different concentrations of diluted CO₂ (77%, 50% and 23%, another mixed gas was N₂), the close-DAE-BPy-CoPor produced CO with FE_{CO} ≥90% under 77% CO₂ + 23% N₂, while at a lower CO₂ concentration (23% CO₂ + 77% N₂), the FE_{CO} of close-DAE-BPy-CoPor can reach 89.2% at -0.6 V (Figure R18).

We have added the Figures R17-18 in the revised Supplementary Information (Page 37, Supplementary Figure 35 and page 38, Supplementary Figure 36) and related description as follows in the revised manuscript (Pages 16-17, lines 395-403).

“Besides, the close-DAE-BPy-CoPor also demonstrated excellent CO selectivity in a CO₂ + N₂ feeding environment, but only H₂ was obtained under only N₂ feeding environment (Supplementary Fig. 35). Under the different concentrations of diluted CO₂ (77%, 50% and 23%, another mixed gas was N₂), the close-DAE-BPy-CoPor produced CO with FE_{CO} ≥90% under 77% CO₂ + 23% N₂, while at a lower CO₂ concentration (23% CO₂ + 77% N₂), the FE_{CO} of close-DAE-BPy-CoPor can reached 89.2% at -0.6 V, showing the potential of close-DAE-BPy-CoPor for practical applications (Supplementary Fig. 36).”

Figure R17. The CO₂RR performance of close-DAE-BPy-CoPor under pure N₂ feeding environment.

Figure R18. The FE_{CO} **a** and j_{CO} **b** of close-DAE-BPy-CoPor under 77% CO₂ + 23% N₂, 50% CO₂ + 50% N₂ and 23% CO₂ + 77% N₂ feeding environment.

3) Revisions

i) The pore size in-text is mentioned in angstroms but in nm in Figure 1, Line 146. The authors should consider using similar units throughout the manuscript.

Response: Thanks for your useful suggestion, we have corrected the word “29.6 Å” to “2.96 nm” as follows (Page 6, line 137) and checked through the manuscript to avoid this problem (Page 8, line 165).

“Thus, one-dimensional (1D) channels were constructed in the well-aligned 2D open-DAE-BPy-CoPor sheets with a theoretical pore size of 2.96 nm, where the distance between adjacent stacking 2D sheet was 4.58 Å (Fig. 1b).”

“Furthermore, the typical-IV adsorption isotherm curves for open-DAE-BPy-CoPor and BPy-CoPor indicated their mesoporous characters, as revealed by the measured pore size of 2.96 nm (Fig. 2c and Supplementary Fig. 5).”

ii) The N₂ sorption diagrams for the important species can be plotted together to effectively visualize the enhancements in the close-DAE-COFs.

Response: As per your suggestion, we have added the plotted the N₂ sorption diagrams of open-DAE-BPy-CoPor, BPy-CoPor and close-DAE-BPy-CoPor together

in the revised Supplementary Information (Page 18, Supplementary Figure 16) and related description as follows in the revised manuscript (Page 12, lines 274-277).

“After photocyclization, the close-DAE-BPy-CoPor still has a high N₂ adsorption uptake with a large BET surface area of 605.2 m² g⁻¹ (Fig. 4b), which was slightly lower than that of open-DAE-BPy-CoPor (S_{BET} = 899.4 m² g⁻¹, Fig. 2c and Supplementary Fig. 16).”

Figure R19. The N₂ sorption isotherms of open-DAE-BPy-CoPor, BPy-CoPor and close-DAE-BPy-CoPor.

iii) The SEM and TEM images (Fig. 2d, 7a-d) are hard to compare with different scales for different species. Would it be possible to employ similar scales?

Response: Thanks for your insightful comment. According to your suggestion, we have tested the SEM of open-DAE-BPy-CoPor and the TEM of BPy-CoPor again to obtain similar scales (nanoscale) SEM and TEM images to convenient for readers. We also repalced Supplementary Figure 7 in the original manuscript with Figure R20 in the revised Supplementary Information (Page 9, Supplementary Figure 7).

Figure R20. The SEM images of **a** BPy-CoPor and **b** open-DAE-BPy-CoPor, and the TEM images of **c** BPy-CoPor and **d** open-DAE-BPy-CoPor.

iv) Fig. 4 c-d does not provide any indication regarding the green and grey lines.

Response: Thanks for your kind suggestion. We have added the details information for green and grey lines in the revised manuscript (Page 10, lines 225-226, Fig. 4).

The grey line was the experimental data and green line was fitting curve.

Fig. 4 The characterization of close-DAE-BPy-CoPor. **a** Comparison of the PXRD

patterns of open-DAE-BPy-CoPor and close-DAE-BPy-CoPor. **b** The N₂ sorption isotherms and pore size distribution of close-DAE-BPy-CoPor. **c** and **d** the X-ray photoelectron spectroscopy of S 2p region for open-DAE-BPy-CoPor and close-DAE-BPy-CoPor, experimental data, grey line; fitting curve, green line.

v) XPS: The relevant information or references that associate the spectral signatures with the respective Co-orbitals should be included.

Response: As per your suggestion, we added the references in the revised manuscript (Page 28, lines 719-724).

“51. Wu, Q.-J., Si, D.-H., Ye, S., Dong, Y.-L., Cao, R., Huang, Y.-B. Photocoupled electroreduction of CO₂ over photosensitizer-decorated covalent organic frameworks. *J. Am. Chem. Soc.* **145**, 19856-19865 (2023).

52. Han, B., et al. Two-dimensional covalent organic frameworks with cobalt(II)-phthalocyanine sites for efficient electrocatalytic carbon dioxide reduction. *J. Am. Chem. Soc.* **143**, 7104-7113 (2021).”

vi) The authors should try to relate the experimental increase in efficiency (1.2-fold increase) to the difference in free energy for the rate-determining step (0.04 eV). The computational energies point towards five times increase in the reaction rate, which agrees with the experimental observations.

Response: Thanks for your suggestion. In this study, the DFT calculation was used to qualitatively investigate the electrocatalytic performance of the close-DAE-BPy-COF. In general, the lower free energy for the rate-determining step of the close-DAE-BPy-COF than that of the open-DAE-BPy-COF, indicating the high CO₂RR performance of close-DAE-BPy-CoPor in the CO₂RR, which is consistent with the CO₂RR experiment.

And the Faradaic efficiency is a parameter used to describe the efficiency of an electrochemical reaction. While it is related to the conversion rate, it cannot straightforwardly reflect changes in conversion rate due to the complex electrocatalytic reaction. Thus, the 1.2-fold increase in CO selectivity is

close-DAE-BPy-CoPor compared to the open-DAE-BPy-CoPor, which should not be simplistically interpreted as a conversion rate of growth.

In the ideal conditional, ΔG is related to the equilibrium constant as follows: $\Delta G = -RT \ln K$ (where R is gas constant, T is temperature, and K is the equilibrium constant). The relationship between equilibrium constant and conversion rate is as follows: $K = \frac{[C]^c [D]^d}{[A]^a [B]^b}$ and $\alpha = \frac{(nA - nB)}{nA}$ (where K is the equilibrium constant, $[A]$, $[B]$, $[C]$, $[D]$ are concentration of reactants and products respectively, a , b , c , d are moles of reactants and products respectively, α is conversion rate, and nA and nB are moles of reactants A and B respectively). Thus, there is a relationship between ΔG and conversion rate in ideal conditions.

However, due to the calculation simplifies the reaction microenvironment, the reaction process cannot be realistically described quantitatively, especially in the complex electrocatalytic reaction. In similar CO₂RR studies, there are no reports to illustrate the relationship between the difference in free energy for the rate-determining step and the experimental increase in efficiency.

vii) The DOS plots must point out the Fermi level. There is no discussion on why there are additional peaks above and below the fermi level in the case of the close-DAE-COF, which are not present in open-DAE-COF. This information may be relevant for further catalyst design.

Response: Thanks for your useful suggestion. As shown in Fig. 6c, we have added the Fermi level (green line). There are additional peaks above and below the Fermi level in the case of the close-DAE-BPy-COF, suggesting more occupied and unoccupied state in its proximity. Thus, the electrons around the Fermi level in close-DAE-BPy-CoPor can transform easily to the unoccupied state than open-DAE-BPy-CoPor under aerobic conditions, which facilitate in enhancing conductivity. (He, X.-G. et. al., *Acta Physica Sinica* 2015, 64, 127301-127301). The assumption was confirmed through experiments conducted by the current density of close-DAE-BPy-CoPor. As shown in Fig. 5f, the j_{CO} of close-DAE-BPy-CoPor can reach up to -20.1 mA cm^{-2} at -1.0 V , which was higher than open-DAE-BPy-CoPor

($j_{CO} = -15.4 \text{ mA cm}^{-2}$) at the same potential under aerobic conditions.

Fig. 6c Projected density of states and integrated density of states of adsorption structures of O₂ on close-DAE-BPy-CoPor and open-DAE-BPy-CoPor.

Fig. 5f The CO partial current density of the BPy-CoPor, close-DAE-BPy-CoPor and open-DAE-BPy-CoPor in the co-feeding CO₂ and 5% O₂

Reviewer #4 (Remarks to the Author):

H.-J. Zhu, et al., reported a reticular chemistry-based electrocatalysts for efficient CO₂ reduction reaction (CO₂RR) under aerobic environments by photoswitching derived oxygen passivation strategies. Direct CO₂RR with flue gas would receive great attention as a practical carbon capture utilization (CCU) approach. Idea of applying photoswitching unit 1,2-Bis(5'-formyl-2'-methylthien-3'-yl)cyclopentene (DAE) to covalent organic framework (COF) is innovative. Comprehensive studies

including material synthesis, characterization, in-situ spectroscopy, CO₂RR performance measurement, and theoretical computation are impressive and supports their conclusion. For the better clarity, this reviewer has several concerns about open/close modes of DAE, which needs to be studied more in detail. I recommend major revision of this manuscript for the publication in Nature Communications. Details as follows:

1. It was explained that 3 h exposure of open-DAE-BPy-CoPor to UV light is required to derive photocyclization and close-DAE-BPy-CoPor. Why does the photoswitching require 3 h for the transition of DAE? If the exposure time becomes shorter or longer, what happens to the openness of DAE-BPy-CoPor? How can we understand the kinetics of photoswitching?

Response: Thanks for your useful comment. The photoswitching of DAE (diarylethene) monomers has been studied intensively. For example, Natalia B. Shustova and co-workers utilized a 2 h UV-irradiation process to synthesize closed-form diarylethene-based metal-organic frameworks (*J. Am. Chem. Soc.* 2019, 141, 5350-5358). In our system, due to the deep coloration of Co-porphyrin, we initially controlled the close form of DAE for 3 h based on material properties to maximize the closure of DAE. Following your suggestion, we conducted the photoswitch experiments with shorter and longer time. As shown in Figure R21, the photocyclization efficiency of DAE-BPy-CoPor after 1 h of UV irradiation was only 6.3%. With extended UV exposure for 5 h, compared to the 3 h duration (30%), there are a modest increase, reaching 32%. Therefore, considering the constraints of economic and time limitations, we have opted to set the photocyclization time for close-DAE-BPy-CoPor at 3 h of UV irradiation.

To understand the kinetics of photoswitching, Cheng-Yong Su and co-workers combine a UV-Vis spectrophotometer with the Lambert-Beer law ($\ln(A_0 - A_\infty)/(A_t - A_\infty) = kt$, where A_0 , A_t , and A_∞ represent the absorbance at the initial, intermediate and final states, t is the radiation time, and k is the rate constant) to study the kinetics of DAE photoswitching (*Chem. Sci.* 2020, 11, 8885-8894). The authors have summarized that the photoswitching efficiency of DAE is influenced by various

structural factors based on the rate constant (k), including the steric structure of the material limits and the linking mode between moieties. We attempted to investigate the photoswitching kinetics of DAE-BPy-CoPor employing the techniques described in the aforementioned research. However, as shown in the Figure R22, due to the partial overlap of absorption peaks in DAE-BPy-CoPor with the close form of DAE, it becomes challenging to elucidate the photoswitching kinetics of DAE-BPy-CoPor. Furthermore, the photoswitching efficiency of DAE is also influenced by factors such as solvent, temperature, solution concentration and irradiation time, as summarized based on previously reported studies.

We have added the Figure R21 in the revised Supplementary Information (Page 14, Supplementary Figure 12) and related description as follows in the revised manuscript (Page 11, lines 242-248).

“Besides, the photocyclization efficiency of DAE is influenced by the duration of UV irradiation exposure, the XPS analysis of close-DAE-BPy-CoPor following 1 h and 5 h after irradiation by UV light revealed the occurrence of 6.3% and 32% photocyclization, respectively (Supplementary Fig. 12). Therefore, considering the constraints of economic and time limitations, we have opted to set the photocyclization time for close-DAE-BPy-CoPor at 3 h of UV irradiation.”

Figure R21. The X-ray photoelectron spectroscopy of S 2p region for close-DAE-BPy-CoPor following 1 h and 5 h after irradiation by UV light.

Figure R22. The liquid UV-Vis absorption spectrum of open-DAE-BPy-CoPor, close-DAE-BPy-CoPor, open-DAE and close-DAE.

2. How can we distinguish the degree of openness of DAE-BPy-CoPor? Although authors provided XPS S 2p (Fig. 4d) and evaluation of O₂-toleration, it is difficult to find a methodology to determine whether all DAEs are fully closed or not.

Response: Thanks for your kind comment. According to published works, we have summarized evaluation methods for assessing the degree of openness or closeness of DAE (diarylethene). Primarily involve the application of X-ray photoelectron spectroscopy (XPS) (Dolgoplova, E. A., *et al.*, *J. Am. Chem. Soc.* 2019, 141, 5350-5358), UV-Vis absorption spectroscopy (Hecht, S. *et al.*, *J. Am. Chem. Soc.* 2015, 137, 2738-2747), and nuclear magnetic resonance (NMR) spectroscopy (Zhu, W.-H. *et al.*, *Chem. Sci.* 2023, 14, 6237-6243). In the DAE-BPy-CoPor, based on the intrinsic properties of the COF, such as its pronounced coloration and insolubility, we are constrained to exclusively employ XPS as the analytical technique for assessing the catalytic photoswitching efficiency. Presently, XPS data reveals that the closed-ring efficiency of close-DAE-BPy-CoPor is 30%, indicating that it has not reached a state of complete ring closure. Due to the presence of isomeric structure in DAE, it can be deduced, based on the Woodward-Hoffmann rules, that its photocyclization efficiency is affected to a certain extent, thus, improving the photocyclization efficiency of DAE is a critical research objective.

3. In the CO₂RR performance results, authors show product distribution of CO in 100% CO₂ and aerobic conditions. I recommend to providing all Faradaic efficiencies (FEs) for CO₂RR and HER such as CO and H₂ FE together.

Response: Thanks for your constructively suggestion. As per your suggestion, we provided the all Faradaic efficiencies of close-DAE-BPy-CoPor under CO₂ and aerobic conditions, as shown in Figures R23 and R24. We added the relevant Figures and information in the revised Supplementary Information (Page 25, Supplementary Figure 23 and page 34, Supplementary Figure 32) and related description as follows in the revised manuscript (Page 14, line 330 and page 16, line 392).

“As shown in Fig. 5b, the close-DAE-BPy-CoPor exhibited remarkable Faradaic efficiencies of CO (FE_{CO}) ($\geq 90\%$) across the entire potential window from at -0.6 V to -0.9 V (vs. RHE), which were larger than those of BPy-CoPor and open-DAE-BPy-CoPor under pure CO₂ gas (Supplementary Fig. 23 and Supplementary Table 4).”

“The optimal FE_{CO} was up to 90.7% at -1.0 V for close-DAE-BPy-CoPor, which was nearly 2.2-fold, 2.2-fold, 3.5-fold and 1.2-fold higher than those of Co-TAPP ($FE_{CO} = 41.1\%$, -1.2 V), COF-366(Co) ($FE_{CO} = 41.0\%$, -1.2 V), BPy-CoPor ($FE_{CO} = 26.0\%$, -1.1 V) and open-DAE-BPy-CoPor ($FE_{CO} = 75.3\%$, -1.1 V) (Supplementary Fig. 32).”

Figure R23. The all Faradaic efficiencies of **a** close-DAE-BPy-CoPor, **b** open-DAE-BPy-CoPor and **c** BPy-CoPor under pure CO₂.

Figure R24. The all Faradaic efficiencies of **a** close-DAE-BPy-CoPor, **b** open-DAE-BPy-CoPor and **c** BPy-CoPor under aerobic conditions. Due to the product of ORR is hard to detect in the aqueous solution, the FE of ORR is additional part of the FE of CO and H₂ ($FE_{ORR} = 100\% - (FE_{CO} + FE_{H_2})$).

4. In the 100% CO₂ condition, except for the potential of -0.5 V (vs. RHE), three samples (Close/Open-DAE-BPy-CoPor, and BPy-CoPor) exhibit similar CO selectivities. Why does BPy-CoPor show degraded CO selectivity compared to others especially at -0.5 V (vs. RHE)?

Response: Thanks for your kind comment. Based on the LSV (Figure R25a), BPy-CoPor exhibits a lower current density compared to the close/open-DAE-BPy-CoPor. Particularly at an applied potential of -0.5 V (vs. RHE), it shows only a total current density of -0.5 mA cm⁻². Consequently, the generated CO concentrations closely approaches of the detection limit of our instrument, leading to significant measurement errors. This phenomenon is also evident from the error bars associated with its CO selectivity (Figure R25b).

Figure R25. a Linear sweep voltammetry curves and b the CO Faradic efficiency of the BPy-CoPor, close-DAE-BPy-CoPor and open-DAE-BPy-CoPor.

5. In the CO₂RR under aeroconditions, I understand the superior CO₂RR performance of the close-BPy-CoPor in Fig. 5e. Although open-BPy-CoPor shows lower CO FE compared to close-BPy-CoPor, BPy-CoPor has a huge drop in CO FE more than open-BPy-CoPor. What is the reason for this phenomenon? Based on operando ATR-FTIR, open-BPy-CoPor also exhibits *OOH under CO₂+O₂ mixed gas condition. I recommend the measurement of operando ATR-FTIR of BPy-CoPor under CO₂+O₂ mixed gas to discover the difference between open-BPy-CoPor and Bpy-CoPor.

Response: Thanks for your constructive suggestion. We believe that the significant

difference in CO selectivity between BPy-CoPor and open-DAE-BPy-CoPor is arisen from their structural disparities, resulting in distinct oxygen activation capabilities. In the absence of DAE species to suppress oxygen activation within the BPy-CoPor, thus, the BPy-CoPor predominantly undergoes oxygen reduction reactions. Therefore, BPy-CoPor exhibited lower CO selectivity under CO₂+O₂ mixed gas condition. Furthermore, as per your suggestion, we added the operando ATR-FTIR of BPy-CoPor under CO₂ + O₂ mixed gas. As shown in Figure R26a, the band located at 1398 cm⁻¹ in the open-BPy-CoPor and BPy-CoPor spectra under co-feeding CO₂ and O₂ was assigned to a carboxyl intermediate of *COOH. More importantly, in the BPy-CoPor curve, an obviously downward broad band at 1637 cm⁻¹ was observed, which was assigned to the H-O-H bending generation. It suggested a hint of water production from the ORR reaction. Furthermore, the intensity of H-O-H bending band is stronger than *COOH band, which indicated the ORR rather than CO₂RR was more easily happened for BPy-CoPor. Thus, BPy-CoPor showed low CO selectivity under CO₂ + O₂ mixed gas (Figure R27).

We have added Figure R26 in the revised Supplementary Information (Page 43, Supplementary Figure 41) and related description as follows in the revised manuscript (Page 17, lines 417-426).

“Furthermore, the operando ATR-FTIR measurement of BPy-CoPor and open-DAE-BPy-CoPor under CO₂ + O₂ mixed gas showed that the band located at 1398 cm⁻¹ in the open-DAE-BPy-CoPor and BPy-CoPor spectra under co-feeding CO₂ and O₂ was assigned to a carboxyl intermediate of *COOH (Supplementary Fig. 41). More importantly, in the BPy-CoPor curve, an obviously downward broad band at 1637 cm⁻¹ was observed, which was assigned to the H-O-H bending generation. It suggested a hint of water production during the reaction. Furthermore, the intensity of H-O-H bending band is stronger than *COOH band, which indicated the ORR rather than CO₂RR was more easily happened for BPy-CoPor.”

Figure R26. **a** Operando ATR-FTIR spectra on BPy-CoPor in the co-feeding CO₂ and 5% O₂ 0.5 M KHCO₃. **b** The comparison operando ATR-FTIR spectra of close-DAE-BPy-CoPor, open-DAE-BPy-CoPor and BPy-CoPor in the co-feeding CO₂ and 5% O₂ 0.5 M KHCO₃.

Figure R27. The CO Faradic efficiency of BPy-CoPor, open-DAE-BPy-CoPor, close-DAE-BPy-CoPor, COF-366(Co) and Co-TAPP under aerobic conditions.

6. Authors mentioned that there was no Co or CoOx based nanoparticle formation after CO₂RR with Supplementary Figs. 22-23, which show TEM images. I recommend investigating XRD and FT-IR to confirm the stability of COF after CO₂RR.

Response: Thanks for your useful suggestion. We added the XRD and FT-IR of

close-DAE-BPy-CoPor after CO₂RR. As shown in Figure R28a, the XRD analysis of the close-DAE-BPy-CoPor revealed an absence of feature peaks associated with Co or CoO_x, while preserving the pristine structure of the COF. Furthermore, the FT-IR of the close-DAE-BPy-CoPor further demonstrated that, in comparison to the pristine one, there are no alteration in the COF's fundamental structure after CO₂RR, confirming COF structural integrity (Figure R28b).

We have added Figure R28 in revised Supplementary Information (Page 33, Supplementary Figure 31) and related description as follows in revised manuscript (Page 16, lines 380-382).

“Furthermore, the PXRD pattern and FT-IR of the close-DAE-BPy-CoPor after CO₂RR remains consistent with that of the as-synthesized COF, confirming the structural integrity of the close-DAE-BPy-CoPor (Supplementary Fig. 31).”

Figure R28. **a** The PXRD pattern close-DAE-BPy-CoPor after CO₂RR. **b** The FT-IR of close-DAE-BPy-CoPor, close-DAE-BPy-CoPor after CO₂RR and carbon black mixed with Nafion.

7. In Supplementary Figure 16a, there is a typo; both black and red denote after states. Furthermore, when we check the difference between before/after of DPBF absorption spectrum, close- and open-DAE BPy-CoPor exhibit 0.093 and 0.11. Are they different enough to see the difference between them?

Response: As per your suggestion, we have corrected the typo in Supplementary Figure 16a in the original Supplementary Information, the black denoted before states,

and we also have checked through the manuscript to avoid this problem. Besides, in our work, we employed DPBF as the probe for detecting reactive oxygen species for COFs due to its exceptionally sensitive to reactive oxygen species, comparing other detections (such as (9,10-anthracenediylbis(methylene)dimalonic acid (ABDA), 9,10-diphenylanthracene (DPA) and sodium 1,3-cyclohexadiene-1,4-diethanoate (CHDDE)). The DPBF exhibits a strong absorption band at 415 nm, and the decrease in adsorption at this wavelength proportional to the amount of the oxygen species generated (Figure R29). This testing method has been employed in an abundant array of previously published articles. Thus, small decay in the absorbance of close-DAE-BPy-CoPor and open-DAE-BPy-CoPor enough to reflect the difference between them. Furthermore, we add the decay rate of close-DAE-BPy-CoPor, open-DAE-BPy-CoPor and BPy-CoPor to compare the difference between them more clearly. As shown in Figure R30, close-DAE-BPy-CoPor showed 12% of DPBF absorption decreased compared to its initial absorption, whereas the open-DAE-BPy-CoPor and BPy-CoPor exhibited an 15% and 19% decrease.

We have added Figure R30 in revised Supplementary Information (Page 22, Supplementary Figure 20) and replace the Supplementary Figure 16a in the original Supplementary Information with Figure R29 in revised Supplementary Information (Page 21, Supplementary Figure 19), and related description as follows in revised manuscript (Page 13, lines 297 to 302).

“As shown in Supplementary Figs. 19 and 20, the absorbance at $\lambda = 410$ nm showed the largest degradation in the BPy-CoPor (0.16, 19% of DPBF absorption decreased), meanwhile, the close-DAE-BPy-CoPor displayed the weakest degradation (0.093, 12% of DPBF absorption decreased), indicating excellent O₂ toleration ability of close-DAE-BPy-CoPor.”

Figure R29. The DPBF absorption spectrum of a BPy-CoPor, b open-DAE-BPy-CoPor and c close-DAE-BPy-CoPor in 0.1 M TBAPF₆/MeCN at -0.1 V vs. Ag/AgCl.

Figure R30. The decay rate of DPBF upon close-DAE-BPy-CoPor, open-DAE-BPy-CoPor and BPy-CoPor.

REVIEWER COMMENTS

Reviewer #1 (Remarks to the Author):

In the revised manuscript, the authors have made a thorough revision following the reviewers' suggestions. The quality and readability of this article are better than the previous one. Most questions are carefully considered and solved. However, there are still issues that the authors should address, which are detailed below.

1. The electrocatalytic performances of open-DAE-BPy-CoPor and close-DAE-BPy-CoPor are different, which suggests that the ring opening/closing exert significant effect on the electronic structures of the metal active centers. Importantly, to reveal such electronic structure regulation, the XANES and EXAFS spectra of both open-DAE-BPy-CoPor and close-DAE-BPy-CoPor, or in situ spectra upon UV irradiation should be conducted and provided.
2. In the EXAFS spectra of open-DAE-BPy-CoPor, there is a strong peak at 2.5 Å. Does it correspond to CoO?
3. The authors attributed the low current densities to the poor CO₂ solubility in H-type cell. However, many works reported high current densities higher than 50 mA/cm². Taking the CO₂ solubility into consideration, the limited CO partial current density in aqueous H-type cell is generally considered to be 70-80 50 mA/cm².

Reviewer #2 (Remarks to the Author):

The authors have addressed my comments. I recommend this manuscript for publication.

Reviewer #3 (Remarks to the Author):

Revised manuscript now addresses the concerns from this reviewer. Especially, their efforts to understand the optimized photoswitching time of DAE, investigate operando ATR-FTIR under CO₂+O₂ mixed gas, characterize the material states of close-DAE-BPy-CoPor after CO₂RR, and investigate the O₂ toleration ability of close-DAE-BPy-CoPor improved the quality of the manuscript. I recommend the acceptance of this manuscript in Nature Communications.

Reviewer #4 (Remarks to the Author):

Reviewer #1 (Remarks to the Author)

In the revised manuscript, the authors have made a thorough revision following the reviewers' suggestions. The quality and readability of this article are better than the previous one. Most questions are carefully considered and solved. However, there are still issues that the authors should address, which are detailed below.

1. The electrocatalytic performances of open-DAE-BPy-CoPor and close-DAE-BPy-CoPor are different, which suggests that the ring opening/closing exert significant effect on the electronic structures of the metal active centers. Importantly, to reveal such electronic structure regulation, the XANES and EXAFS spectra of both open-DAE-BPy-CoPor and close-DAE-BPy-CoPor, or in situ spectra upon UV irradiation should be conducted and provided.

Response: We highly appreciate the suggestions to improve the quality of our paper. As per your suggestion, we tried our best to test the X-ray absorption spectra of open-DAE-BPy-CoPor and close-DAE-BPy-CoPor to illustrate the electronic structure on the CoN₄ centers. As shown in Figure R1a, the Co K-edge XANES profile of close-DAE-BPy-CoPor exhibits a similar wave feature with that of open-DAE-BPy-CoPor, indicating that the Co center valence of close-DAE-BPy-CoPor similar with open-DAE-BPy-CoPor. Notably, compared with open-DAE-BPy-CoPor, the absorption peak of close-DAE-BPy-CoPor at around 7715 eV shifted to the lower-energy side. This shift implies a slightly lower Co oxidation state in close-DAE-BPy-CoPor than open-DAE-BPy-CoPor, indicating the presence of more electrons in the Co center of close-DAE-BPy-CoPor. Moreover, the Fourier-transformed Co K-edge EXAFS spectrum for open-DAE-BPy-CoPor shows a dominant peak at 1.44 Å, which could be attributed to the Co-N bond. Compared with open-DAE-BPy-CoPor, the intensity of Co-N peak for close-DAE-BPy-CoPor is negatively shifted ($\Delta = 0.02$ Å), indicating a reduced bond length of Co-N in close-DAE-BPy-CoPor (Figure R1b). The EXAFS fitting results further prove that the bond length of Co-N in close-DAE-BPy-CoPor (1.93 Å) is shorter than that of Co-N in open-DAE-BPy-CoPor (1.95 Å) (Table R1). The contraction of the Co-N bond

length can facilitate in transfer electrons to the Co center, indicating that the Co center electrons of close-DAE-BPy-CoPor is more than open-DAE-BPy-CoPor. This phenomenon is also found in other catalyst system, such as the Hg-CoTPP (Wai-Yeung Wong et al. *J. Am. Chem. Soc.* 2022, 144, 15143-15154) and NiPc-CN molecularly dispersed electrocatalysts (Yongye Liang et al. *Nat. Energy.* 2020, 5, 684-692).

Although we strongly hope to test in situ spectra as the Reviewer suggestion, after discuss with Prof. Qing Xu in Shanghai Institute of Advanced Research and Shanghai Synchrotron Radiation Facility, there exists technical challenges for the in situ XANES and EXAFS spectra under UV irradiation. Therefore, we adopted a semi-in situ XANES and EXAFS detection method to systematically investigate the electronic structure changes at the Co center of DAE-BPy-CoPor under varying UV irradiation time (1 h, 3 h and 6 h). As shown in Figure R2a, the Co K-edge XANES profile of DAE-BPy-CoPor under varying UV irradiation time (1 h, 3 h and 6h) also exhibits a similar wave feature with that of open-DAE-BPy-CoPor. However, the absorption peak consistently shows a gradual shift to the lower-energy side as the UV irradiation time increases. Moreover, in comparison to open-DAE-BPy-CoPor, the EXAFS spectra of close-DAE-BPy-CoPor reveal a consistent negative shift in the Co-N peak as the duration UV irritation varies (Figure R2b). These results suggest that the electronic structures of the Co centers in DAE-BPy-CoPor are affected by the photocyclization efficiency of DAE monomers. It indicated that the presence of more electrons in the Co center of DAE-BPy-CoPor with the increase of the degree of cyclization, which would facilitate in the activation and reduction of CO₂.

We have added the Figures R1-R2 and Table R1 in the revised Supplementary Information (Page 18, Supplementary Figure 16; Page 19, Supplementary Figure 17; page 60, Supplementary Table 3) and related description (Pages 11-12, lines 264-292) and references (Page 29-30, lines 757-763) as follows in the revised manuscript.

“Furthermore, the Co K-edge XANES profile of close-DAE-BPy-CoPor exhibited a similar wave feature with that of open-DAE-BPy-CoPor, indicating that the Co center

valence of close-DAE-BPy-CoPor similar with open-DAE-BPy-CoPor. Notably, compared with open-DAE-BPy-CoPor, the absorption peak of close-DAE-BPy-CoPor at around 7715 eV shifted to the lower-energy side. This shift implied a slightly lower Co oxidation state in close-DAE-BPy-CoPor than open-DAE-BPy-CoPor, indicating the presence of more electrons in the Co center of close-DAE-BPy-CoPor. Moreover, the Fourier-transformed Co K-edge EXAFS spectrum for open-DAE-BPy-CoPor showed a dominant peak at 1.44 Å, which could be attributed to the Co-N bond. Compared with open-DAE-BPy-CoPor, the intensity of Co-N peak for close-DAE-BPy-CoPor was negatively shifted ($\Delta = 0.02$ Å), indicating a reduced bond length of Co-N in close-DAE-BPy-CoPor (Supplementary Fig. 16b). The EXAFS fitting results further proved that the bond length of Co-N in close-DAE-BPy-CoPor (1.93 Å) was shorter than that of Co-N in open-DAE-BPy-CoPor (1.95 Å) (Supplementary Table 3). The contraction of the Co-N bond length can facilitate in transfer electrons to the Co center^{56,57}, indicating that the Co center electrons of close-DAE-BPy-CoPor was more than open-DAE-BPy-CoPor. Furthermore, the Co K-edge XANES profile of DAE-BPy-CoPor under varying UV irradiation time (1 h, 3 h and 6h) also exhibited a similar wave feature with that of open-DAE-BPy-CoPor. However, the absorption peak consistently showed a gradual shift to the lower-energy side as the UV irradiation time increased (Supplementary Fig. 17a). Moreover, in comparison to open-DAE-BPy-CoPor, the EXAFS spectra of close-DAE-BPy-CoPor reveal a consistent negative shift in the Co-N peak as the duration UV irritation varies (Supplementary Fig. 17b). These results suggested that the electronic structures of the Co centers in DAE-BPy-CoPor were affected by the photocyclization efficiency of DAE monomers. It indicated that the presence of more electrons in the Co center of DAE-BPy-CoPor with the increase of the degree of cyclization, which would facilitate in the activation and reduction of CO₂.”

“56. Fang, M., Xu, L., Zhang, H., Zhu, Y., Wong, W.-Y. Metalloporphyrin-linked mercurated graphynes for ultrastable CO₂ electroreduction to CO with nearly 100% selectivity at a current density of 1.2 A cm⁻². J. Am. Chem. Soc. **144**, 15143-15154

(2022).

57. Zhang, X., et al. *Molecular engineering of dispersed nickel phthalocyanines on carbon nanotubes for selective CO₂ reduction. Nat. Energy* 5, 684-692 (2020).”

Figure R1. The local coordination structure. **a** Co *K*-edge of X-ray absorption near-edge structure spectra of open-DAE-BPy-CoPor and close-DAE-BPy-CoPor. **b** Co *K*-edge of EXAFS spectra of open-DAE-BPy-CoPor and close-DAE-BPy-CoPor. **c** The extended X-ray absorption fine structure fitting curves of close-DAE-BPy-CoPor.

Figure R2. The local coordination structure of open-DAE-BPy-CoPor and close-DAE-BPy-CoPor under varying durations of UV irradiation (1 h, 3 h and 6h). **a** Co *K*-edge of X-ray absorption near-edge structure spectra. **b** Co *K*-edge of EXAFS spectra.

Table R1. Fitting results from EXAFS analysis of open-DAE-BPy-CoPor. (CN: coordination number; R: distance between absorber and backscatter atoms; σ^2 : Debye-Waller factor (a measure of thermal and static disorder in absorber-scatterer distances); ΔE_0 : the inner potential correction; R factor is used to value the goodness of the fitting.)

Sample	Path	CN	R(Å)	$\sigma^2(10^{-3} \text{ \AA}^2)$	ΔE_0 (eV)	R factor
Open-DAE-BPy-CoPor	Co-N	4.3 \pm 0.3	1.95	6.32	-6.48	0.02
Close-DAE-Bpy-CoPor	Co-N	4.0 \pm 0.6	1.93	4.97	-1.21	0.003

2. In the EXAFS spectra of open-DAE-BPy-CoPor, there is a strong peak at 2.5 Å. Does it correspond to CoO?

Response: Thanks for your kind comments. Upon careful examination of the peak of open-DAE-BPy-CoPor at 2.5 Å, which the reviewer initially associated with the Co-Co peak in the R space of CoO, we observed that this peak is actually offset from the CoO (Figure R3). But it aligns consistently with the standard sample Co-TPPCN molecule (only Co-N₄ structure and without CoO). Furthermore, according to reported works (*Nat. Commun.*, **2023**, 14, 3401; *Nat. Commun.* **2020**, 11, 4173), it has been established that the peak at 2.5 Å does not correspond to CoO. In addition, we added wavelet transform (WT) to prove that there is no CoO species existed in open-DAE-BPy-CoPor. As shown in Figure R4, compared to the WT contour plots of CoO with the feature of Co-Co coordination, the band corresponding to Co-N shown in the WT contour plots of open-DAE-BPy-CoPor and Co-TPPCN without the feature of Co-Co coordination, indicating the absence of CoO species in open-DAE-BPy-CoPor. Moreover, the absence of CoO characteristics in both PXRD (Figure R5), TEM, HR-TEM and EDX elemental mapping (Figure R6) tests.

We have added the Figures R4 in the revised Supplementary Information (Page 13, Supplementary Figure 11) and related description (Pages 9-10, lines 216-221) as follows in the revised manuscript.

“In addition, the wavelet transform (WT) analysis has been conducted to prove that there was no CoO species existed in open-DAE-BPy-CoPor. As shown in Supplementary Fig. 11, compared to the WT contour plots of CoO with the feature of Co-Co coordination, the band corresponding to Co-N shown in the WT contour plots of open-DAE-BPy-CoPor and Co-TPPCN without the feature of Co-Co coordination, indicating the absence of CoO species in open-DAE-BPy-CoPor.”

Figure R3. Co *K*-edge of EXAFS spectra of open-DAE-BPy-CoPor, Co-TPPCN and CoO.

Figure R4. Wavelet transformed (WT) EXAFS analysis plots of open-DAE-BPy-CoPor, Co-TPPCN and CoO.

Figure R5. The PXRD patterns of open-DAE-BPy-CoPor and CoO.

Figure R6. Transmission electron microscopy **a** and **b** high resolution transmission electron microscopy images. **c** Aberration-corrected high-angle annular dark-field scanning transmission electron microscopy image and energy-dispersive X-ray spectroscopy elemental mapping patterns of open-DAE-BPy-CoPor.

3. The authors attributed the low current densities to the poor CO₂ solubility in H-type cell. However, many works reported high current densities higher than 50 mA/cm². Taking the CO₂ solubility into consideration, the limited CO partial current density in aqueous H-type cell is generally considered to be 70-80 mA/cm².

Response: Thanks for your useful comments. As shown in Table R2, we have summarized the reported electrocatalysts for CO₂RR in H-type cell. The CO partial current density of close-DAE-BPy-CoPor in CO₂-saturated electrolyte shows -8.5 mA cm⁻² at -0.9 V which is higher than most of reported cobalt porphyrin-based electrocatalysts including BPy-CoPor, ViB12@rGO, Co-TTCOF and COF-366-Co. But, the current densities of close-DAE-BPy-CoPor still lower than Ni-N₃-V, C-Zn₁-Ni₄ ZIF-8, rGO-PEI-MoS_x and CoPcPDQ-COF with excellent conductivity. Based on those electrocatalysts with high current densities, we find that the higher current density of CO is not solely dependent on the previously discussed CO₂ solubility, but also being related with electrical conductivity. The inefficiency in the utilization of the catalytic active center may attributed to the limited contact between COFs in CO₂RR, which could be a factor to the lower current density observed in our catalyst within the H-type cell. Therefore, we increased the amount of carbon black from 0.5 times (carbon black : electrocatalyst) to 1.5 times. As shown in Figure R7a,

the linear sweep voltammetry (LSV) curves of carbon black/close-DAE-BPy-CoPor (1.5 : 1) shows higher current densities than carbon black/close-DAE-BPy-CoPor (0.5 : 1). This result suggests that the increased current may be attributed to the enhanced contact between COFs facilitated by carbon black, leading to improved utilization of catalytic sites. The current density of carbon black/close-DAE-BPy-CoPor (1.5 : 1) can reach -47.4 mA cm^{-2} at -1.2 V vs. RHE, which was almost 1.5-fold than carbon black/close-DAE-BPy-CoPor (0.5 : 1) (-31.3 mA cm^{-2}). The carbon black/close-DAE-BPy-CoPor (1.5 : 1) also exhibited higher Faradaic efficiencies of CO (FE_{CO}) ($\geq 80\%$) across the entire potential window from at -0.6 V to -1.2 V (Figure R7b). Particularly, the CO partial current densities of carbon black/close-DAE-BPy-CoPor (1.5 : 1) achieved -38.0 mA cm^{-2} at -1.2 V (Figure R7c). Besides, only H_2 was obtained on the carbon paper only coated with carbon black in CO_2 -saturated 0.5 M KHCO_3 (Figure R8).

We have added the Figures R7–R8 and updated Supplementary Table 4 to Table R2 in the revised Supplementary Information (Page 29, Supplementary Figure 27; page 30, Supplementary Figure 28) and related description (Pages 15, lines 356-361) as follows in the revised manuscript.

“In addition, with increasing the amount of carbon black from 0.5 times (carbon black : electrocatalyst) to 1.5 times, the j_{CO} of close-DAE-BPy-CoPor achieved -38.0 mA cm^{-2} at -1.2 V (Supplementary Fig. 27). Besides, only H_2 was obtained on the carbon paper only coated with carbon black in CO_2 -saturated 0.5 M KHCO_3 (Supplementary Fig. 28).”

Table R2. The summary of CO_2 electroreduction performances for reported electrocatalysts and this work.

Catalyst	electrolyte	Highest FE_{CO} (%)	j_{CO} (mA cm^{-2})	Stability (h)	Ref.
Close-DAE-BPy-CoPor	0.5 M KHCO_3	98.0	-8.47 (-0.9 V)	24	This work
Carbon black/Close-DAE-BPy-CoPor	0.5 M KHCO_3	80.2	-38.0 (-1.2 V)	NA	This work

(1.5 : 1)					
Open-DAE-BPy-CoPor	0.5 M KHCO ₃	95.2	-6.99 (-0.9 V)	NA	This work
BPy-CoPor	0.5 M KHCO ₃	94.1	-5.30 (-0.9 V)	NA	This work
ViB ₁₂ @rGO	0.5 M KHCO ₃	94.5	-6.24 (-0.8 V)	10	ACS Appl Mater Interfaces 12, 41288-41293 (2020)
Co-TTCOF	0.5 M KHCO ₃	91.3	-1.84 (-0.7 V)	40	Nat Commun 11, 497 (2020)
COF-366-Co	0.5 M KHCO ₃	90.0	-1.8 (-1.1 V)	24	Science 349, 1208 (2015)
COF-367-Co	0.5 M KHCO ₃	91.0	-3.3 (-1.1 V)	24	Science 349, 1208 (2015)
TTF-Por(Co)-COF	0.5 M KHCO ₃	70.0	-6.88 (-0.9 V)	10	ACS Energy Letters 5, 1005-1012 (2020)
Co-TPP-cov	0.5 M KHCO ₃	67	-1.065 (-0.63 V)	4	Angew. Chem. Int. Ed. 56, 6468-6472 (2017)
Co-TPP/CNT	0.5 M KHCO ₃	91	-3.2 (-0.66 V)	12	Angew. Chem. Int. Ed. 58, 6595-6599 (2019)
Co-Bpy-COF-Ru1/2	0.5 M KHCO ₃	96.7	about -9 (-0.7 V)	13	J Am Chem Soc. (2023)
Ni-N ₃ -V	0.5 M KHCO ₃	94.0	-48 (-0.8 V)	14	Angew. Chem., Int. Ed. 2019 , 59, 1961-1965
C-Zn ₁ -Ni ₄ ZIF-8	0.5 M KHCO ₃	98.0	-55 (-0.8 V)	12	Energy Environ. Sci. 2018 , 11, 1204-1210
rGO-PEI-MoS _x	0.5 M NaHCO ₃	85.1	-55 (-0.65 V)	3	Energy Environ. Sci. 2016 , 9, 216
CoPcPDQ-COF	0.5 M KHCO ₃	96.0	-49.4 (-0.66 V)	24	Angew. Chem. Int. Ed. 2020 , 59, 16587-16593

Figure R7. **a** Linear sweep voltammetry curves of carbon black/close-DAE-BPy-CoPor (0.5 : 1) and carbon black/close-DAE-BPy-CoPor (1.5 : 1), **b** the CO Faradic efficiency and **c** the CO partial current density of carbon black/close-DAE-BPy-CoPor (1.5 : 1).

Figure R8. The CO₂RR performance of carbon black in the CO₂-saturated 0.5 M KHCO₃.

Reviewer #2 (Remarks to the Author):

The authors have addressed my comments. I recommend this manuscript for publication.

Response: Thank you very much for your positive comments.

Reviewer #3 (Remarks to the Author):

Revised manuscript now addresses the concerns from this reviewer. Especially, their efforts to understand the optimized photoswitching time of DAE, investigate operando ATR-FTIR under CO₂+O₂ mixed gas, characterize the material states of

close-DAE-BPy-CoPor after CO₂RR, and investigate the O₂ toleration ability of close-DAE-BPy-CoPor improved the quality of the manuscript. I recommend the acceptance of this manuscript in Nature Communications.

Response: Thank you very much for positive comments.

Reviewer #4 (Remarks to the Author):

Response: Thank you very much for your positive comments.

REVIEWERS' COMMENTS

Reviewer #1 (Remarks to the Author):

The authors have addressed my concerns and thus I recommend the publication of this manuscript as it is.